# Deep Taxonomic Networks for Unsupervised Hierarchical Prototype Discovery

**Zekun Wang**[1], **Ethan Haarer**[1], **Tianyi Zhu**[1,2], **Zhiyi Dai**[1], **Christopher MacLellan**[1]
[1]Georgia Institute of Technology    [2]University of Virginia
{zekun, ehaarer3, zdai83, cmaclell}@gatech.edu
crv5ns@virginia.edu

## Abstract

Inspired by the human ability to learn and organize knowledge into hierarchical taxonomies with prototypes, this paper addresses key limitations in current deep hierarchical clustering methods. Existing methods often tie the structure to the number of classes and underutilize the rich prototype information available at intermediate hierarchical levels. We introduce deep taxonomic networks, a novel deep latent variable approach designed to bridge these gaps. Our method optimizes a large latent taxonomic hierarchy, specifically a complete binary tree structured mixture-of-Gaussian prior within a variational inference framework, to automatically discover taxonomic structures and associated prototype clusters directly from unlabeled data without assuming true label sizes. We analytically show that optimizing the ELBO of our method encourages the discovery of hierarchical relationships among prototypes. Empirically, our learned models demonstrate strong hierarchical clustering performance, outperforming baselines across diverse image classification datasets using our novel evaluation mechanism that leverages prototype clusters discovered at all hierarchical levels. Qualitative results further reveal that deep taxonomic networks discover rich and interpretable hierarchical taxonomies, capturing both coarse-grained semantic categories and fine-grained visual distinctions.

## 1 Introduction

The human mind possesses an extraordinary capacity to learn, organize knowledge, and generalize from experience, often constructing rich, abstract hierarchical category structures [42]. This learning journey begins early; even pre-linguistic infants demonstrate an ability to group objects into rudimentary categories based on salient perceptual features like shape and parts, forming the initial scaffolding of a hierarchical taxonomy without the need for explicit semantic symbols [35, 9, 2]. Two key principles appear fundamental to this organization: the formation of hierarchical taxonomies and the representation of categories via prototypes [35]. We naturally structure our knowledge in nested levels of abstraction (e.g., *collie* → *dog* → *mammal* → *animal*) [37]. Within these hierarchies, a 'basic level' (e.g., *dog*, *chair*) emerges as psychologically privileged [4], representing an optimal trade-off between informativeness and cognitive effort. This prototype serves as a cognitive reference point, allowing for graded membership (e.g., a robin is a more prototypical bird than a penguin) and facilitating efficient generalization to novel instances based on similarity to the prototype [7, 6].

Inspired by these powerful human capabilities, early computational approaches, such as Cobweb [7, 13], explicitly attempted to support unsupervised, incremental learning of hierarchical, probabilistic prototypes by optimizing *category utility*—a measure reflecting the trade-off between feature predictability within a category and distinctiveness between categories [16, 4]. Modern deep learning systems have revisited these themes, developing methods for hierarchical clustering [30, 48, 29], often

integrating hierarchical structures and prototypes within neural networks to enhance performance, interpretability, and robustness. Despite this progress in deep learning, two significant gaps persist. Firstly, existing deep hierarchical clustering approaches often tie leaf nodes to fixed class labels and require retraining to handle different classification granularity on the same data. Secondly, by treating leaf clusters as terminal representations, current approaches overlook intermediate prototypes and underutilize rich multi-level abstractions.

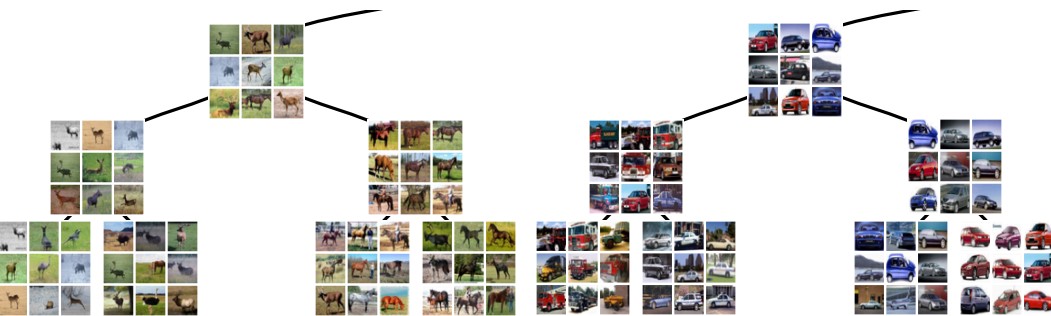

(a) Sub-hierarchy of ungulate. Left branch: deer-like silhouettes (including ostriches). Right branch: grazing versus ridden horses. The parent blends both.

(b) Sub-hierarchy of cars. Parent: blends red, white, and blue cars. Left branch: emergency (red fire trucks, police cars); Right branch: red hatchbacks, blue sedans.

Figure 1: Examples of sub-hierarchies discovered by fitting a deep taxonomic network to CIFAR-10 data. For each cluster, we sampled nine images from the test set based on likelihood.

To bridge this gap, we propose deep taxonomic networks, a novel deep latent variable approach that leverages variational inference [36, 18]. Our approach optimizes a large latent taxonomic hierarchy structured as a complete binary-tree mixture-of-Gaussian prior. This hierarchical prior enables our method to automatically discover abstraction structures and their associated prototypes directly from unlabeled data (see Figure 1). We contribute the following: **(a)** A deep latent variable approach that discovers fine-grained hierarchies from unlabeled data without assuming label counts; **(b)** A theoretical analysis showing that maximizing the ELBO in our approach results in good hierarchical prototypes that describe the data; **(c)** A simple training framework that does not require specialized training procedures or pre-training and fine-tuning, and can seamlessly incorporate contrastive learning [3] jointly with the variational inference objective; **(d)** Our models outperform related hierarchical clustering models on image classification datasets of varying complexity and class labels by a large margin using a novel evaluation mechanism that leverages rich prototypes discovered at all hierarchical levels; **(e)** Our approach discovers rich, interpretable hierarchical prototypes at different granularity.

## 2 Related work

**Hierarchical clustering** Hierarchical clustering organizes data into a nested structure of clusters, revealing relationships at multiple granularities [40, 45]. Traditional agglomerative methods, such as Ward's minimum variance approach, iteratively merges the pairs of clusters by minimizing the increase in total within-cluster variance. [45]. Deep learning based approaches learn the hierarchical clusterings in an embedding space, with the advantage of integrating deep representation learning techniques such as contrastive learning [3] in the clustering framework [30, 29, 48] for more robust performance on high-dimensional data. For example, DeepECT [30] couples an autoencoder with a projection-based divisive clustering layer to recursively split data into a binary tree in the learned embedding space. Yet, many existing methods are limited by fixed class-based hierarchies and an over-reliance on leaf clusters, ignoring rich intermediate prototypes. Most related to our work is Cobweb [7, 26, 1], a concept formation system that builds concept hierarchies top-down based on *categorical utility* [4] and identifies meaningful *basic-level* categories at intermediate nodes. Cobweb is not constrained by label size, and it leverages its entire learned hierarchical structure at inference time, including intermediate prototypes [1]. While Cobweb processes raw inputs and assumes conditional independence of features, deep taxonomic networks employ neural networks to optimize a potentially large, pre-defined taxonomic structure within an embedding space to jointly learn robust data representations and informative hierarchical prototypes at all levels of the tree, facilitating the discovery of *basic-level* clusters.

**Deep latent variable models** Variational autoencoders (VAEs) [36, 18] are deep latent variable approaches that use neural networks (encoder networks $q_\phi(\mathbf{z}|\mathbf{x})$ and decoder networks $p_\theta(\mathbf{x}|\mathbf{z})$) to learn data distributions by optimizing the Evidence Lower Bound (ELBO) objective. In this framework, the prior distribution $p(\mathbf{z})$ represents the underlying observation generation process. While standard VAEs use a standard Gaussian prior $p(\mathbf{z})$, the prior can be adjusted to account for specific structures in the data. VaDE [15] proposes a Gaussian mixture prior to jointly learn latent representations and cluster assignments. Other work [10, 38] leverages a nested Chinese Restaurant Process prior [11] to learn hierarchical latent representations. Alternatively, hierarchical VAEs [19, 27, 44] employ multiple stochastic latent layers to learn multiple approximated posteriors at varying abstraction levels [41, 47]. For instance, MF-VAE [5] uses VLAE [47] to learn multi-faceted clusterings of data, and TreeVAE [29] constructs a tree-like approximated posteriors using LadderVAE [41] for hierarchical clustering. However, these approaches often increase computational cost by optimizing multiple decoder networks and require specialized procedures [5] or frequent fine-tunings [29]. Contrary to these approaches, deep taxonomic networks utilize a complete binary tree mixture-of-Gaussians prior to explicitly support taxonomy within a single approximated posterior.

## 3 Deep Taxonomic Networks

We introduce deep taxonomic networks, a novel deep latent variable approach featuring a complete binary tree Mixture-of-Gaussians prior. This method learns a hierarchical taxonomy by mapping data to the most prototypical clusters, parameterized by their Gaussian priors. These clusters are optimized for high intra-category similarity (low internal feature entropy) and high information gain about the features from cluster membership. We start by describing the generative process within the VAE framework in Section 3.1. Then we describe the variational inference process in Section 3.2 and its connection to prototypicality maximization in Section 3.3. Finally, we introduce a contrastive learning extension for real-world images in Section 3.4.

### 3.1 Generative process with a hierarchical mixture-of-Gaussians prior

We define the conditional prior distribution over the latent variable $\mathbf{z}$ using a hierarchical structure $\mathcal{T}$, represented as a complete binary tree. Each node $c$ in $\mathcal{T}$, has an associated prior probability $p(c)$ and corresponds to a cluster defined by parameters $(\mu_c, \sigma_c^2)$, such that the conditional latent prior $p(\mathbf{z} \mid c) = \mathcal{N}(\mathbf{z} \mid \mu_c, \sigma_c^2\mathbf{I})$. We assume an isotropic Gaussian for simplicity, where $\sigma_c^2\mathbf{I}$ denotes a diagonal covariance matrix. Though our current analysis utilizes a simplified covariance structure, the underlying method is not limited to this configuration and can be extended to other covariance structures. To enforce hierarchical dependency, the parameters of a parent node $c_{\text{parent}}$ are the convex combinations of those of its children, $c_{\text{left}}$

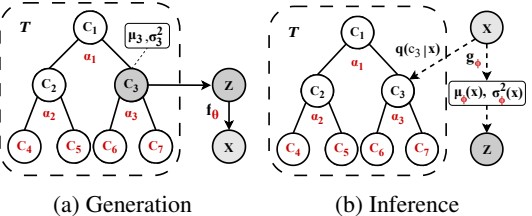

(a) Generation  (b) Inference

Figure 2: The graphic model for deep taxonomic networks. (a): solid arrows represent the generative sampling process. The grayed cluster $c_3$ is selected via the prior distribution $p(c)$. (b): dashed arrows represent the variational inference process. Red: learnable parameters.

and $c_{\text{right}}$. Specifically, we model the distribution at the parent node as a Gaussian approximation to the mixture of its children's distributions. By matching the first two moments of the mixture $\alpha\mathcal{N}(\mu_{c_{\text{left}}}, \sigma_{c_{\text{left}}}^2\mathbf{I}) + (1-\alpha)\mathcal{N}(\mu_{c_{\text{right}}}, \sigma_{c_{\text{right}}}^2\mathbf{I})$, where $\alpha \in [0, 1]$ is a convex weight, we obtain:

$$\mu_{c_{\text{parent}}} = \alpha\mu_{c_{\text{left}}} + (1-\alpha)\mu_{c_{\text{right}}}$$

$$\sigma_{c_{\text{parent}}}^2 = \alpha\left(\sigma_{c_{\text{left}}}^2 + \frac{1}{D}\|\mu_{c_{\text{left}}} - \mu_{c_{\text{parent}}}\|_2^2\right) + (1-\alpha)\left(\sigma_{c_{\text{right}}}^2 + \frac{1}{D}\|\mu_{c_{\text{right}}} - \mu_{c_{\text{parent}}}\|_2^2\right)$$

where $D$ is the dimension of the latent space. These constraints ensure that parent clusters represent broader distributions encompassing their children in the latent space. This approach also reduces the number of learnable parameters as the intermediate clusters are inferred from the leaf clusters and the convex weights at each branch (Figure 2). As shown in Figure 2a, the overall generative process for an observation $\mathbf{x}$ proceeds as follows: (1) select a cluster $c$ in $\mathcal{T}$ via a prior $p(c)$; (2) sample a latent representation $\mathbf{z}$ from the chosen cluster's distribution: $\mathbf{z} \sim \mathcal{N}(\mu_c, \sigma_c^2\mathbf{I})$; (3) generate the observation

$\mathbf{x}$ conditioned on the latent representation via a decoder network $f_\theta$: $\mathbf{x} \sim p_\theta(\mathbf{x} \mid \mathbf{z})$. The decoder defines Gaussian $\mathcal{N}(\mathbf{x} \mid f_\theta(\mathbf{z}), \mathbf{I})$ with unit variance for real-valued $\mathbf{x}$ or Bernoulli for binary $\mathbf{x}$.

## 3.2 Variational inference

Deep taxonomic networks represent the data distribution $p(\mathbf{x})$ using a hierarchical mixture-of-Gaussians prior over $\mathcal{T}$. We achieve this by maximizing the Evidence Lower Bound (ELBO) on $\log p(\mathbf{x})$ using an amortized variational posterior $q_\phi(\mathbf{z}, c \mid \mathbf{x}) = q_\phi(\mathbf{z} \mid \mathbf{x})q_\phi(c \mid \mathbf{x})$, where we parameterize $q_\phi(\mathbf{z} \mid \mathbf{x}) = \mathcal{N}(\mathbf{z} \mid \mu_\phi(\mathbf{x}), \sigma_\phi^2(\mathbf{x})\mathbf{I})$ by an encoder neural network $g_\phi(\mathbf{x})$. The overall inference process is shown in Figure 2b. Formally:

$$\begin{aligned}
\log p(\mathbf{x}) &= \log \int_{\mathbf{z}} \sum_{c \in \mathcal{T}} p_\theta(\mathbf{x} \mid \mathbf{z})p_\theta(\mathbf{z} \mid c)p_\theta(c)d\mathbf{z} \\
&= \log \int_{\mathbf{z}} \sum_{c \in \mathcal{T}} q_\phi(\mathbf{z}, c \mid \mathbf{x})\frac{p_\theta(\mathbf{x} \mid \mathbf{z})p_\theta(\mathbf{z} \mid c)p_\theta(c)}{q_\phi(\mathbf{z}, c \mid \mathbf{x})}d\mathbf{z} \\
&= \log \mathbb{E}_{q_\phi(\mathbf{z}, c \mid \mathbf{x})}\left[\frac{p_\theta(\mathbf{x} \mid \mathbf{z})p_\theta(\mathbf{z} \mid c)p_\theta(c)}{q_\phi(\mathbf{z}, c \mid \mathbf{x})}\right] \geq \mathbb{E}_{q_\phi(\mathbf{z}, c \mid \mathbf{x})}\left[\log \frac{p_\theta(\mathbf{x} \mid \mathbf{z})p_\theta(\mathbf{z} \mid c)p_\theta(c)}{q_\phi(\mathbf{z}, c \mid \mathbf{x})}\right]
\end{aligned}$$
(1)

where right-hand side of Equation (1) represents Jensen's inequality and is the evidence lower bound (ELBO) [18, 36], $\mathcal{L}_{ELBO}(\phi, \theta)$, and can be rewritten as (see full derivation in Appendix A.1):

$$\mathcal{L}_{ELBO}(\phi, \theta) = \mathbb{E}_{q_\phi(\mathbf{z} \mid \mathbf{x})}\left[\log p_\theta(\mathbf{x} \mid \mathbf{z})\right] \tag{2}$$

$$- \mathbb{E}_{q_\phi(c \mid \mathbf{x})}D_{KL}\left(q_\phi(\mathbf{z} \mid \mathbf{x}) \mid\mid p_\theta(\mathbf{z} \mid c)\right) \tag{3}$$

$$- D_{KL}\left(q_\phi(c \mid \mathbf{x}) \mid\mid p_\theta(c)\right) \tag{4}$$

The ELBO objective can be interpreted as follows: Equation (2) measures the reconstruction quality between the encoder $g_\phi(\mathbf{x})$ and the decoder $f_\theta(\mathbf{z})$. Equation (3) is the Kullback–Leibler (KL) divergence between the learned latent distribution of input $\mathbf{x}$ and the clusters in $\mathcal{T}$, weighted by $q_\phi(c \mid \mathbf{x})$, which represents the cluster assignment probability of $\mathbf{x}$. Equation (4) regularize the cluster assignment probability to be close to the prior distribution of the clusters in $\mathcal{T}$. As suggested in [15, 5], we can replace $q_\phi(c \mid \mathbf{x})$ with $p_\theta(c \mid \mathbf{z}) = \frac{p_\theta(c)p_\theta(\mathbf{z} \mid c)}{\sum_{c' \in \mathcal{T}} p_\theta(c')p_\theta(\mathbf{z} \mid c')}$, the cluster assignment probability of $\mathbf{z}$, with one Monte Carlo sample via a reparameterization trick [18].

## 3.3 Reinterpretation of ELBO as prototypicality maximization

In hierarchical clustering approaches, the concept of prototypicality quantifies how well a cluster $c$ within the hierarchy $\mathcal{T}$ encapsulates and informs about a given sample's latent representation $\mathbf{z}$. Categorical utility (CU) [16, 4] formalizes this notion by providing a principled approach from information-theory: let $\mathcal{Z}$ be the random variable over latent vectors $\mathbf{z}$, CU is then defined as the mutual information between $\mathcal{Z}$ and $\mathcal{T}$:

$$\text{CU} = I(\mathcal{Z}; \mathcal{T}) = H(\mathcal{Z}) - \sum_{c \in \mathcal{T}} p(c)H(\mathcal{Z} \mid T = c),$$

$H(\mathcal{Z})$ is the entropy of the feature distribution. $H(\mathcal{Z} \mid \mathcal{T} = c)$ is the conditional entropy of the features given the cluster assignment $c$. CU quantifies the information gained about the features from knowing $c$, favoring clusters in the hierarchical level with low internal entropy $H(\mathcal{Z} \mid T = c)$ and large reduction from $H(\mathcal{Z})$, analogous to cognitive *basic-level* categories [16, 4]. The prototypical cluster $c^* = \arg\max_c[H(\mathcal{Z}) - H(\mathcal{Z} \mid \mathcal{T} = c)]$ optimally balances cue validity (rich, predictable internal features) with category validity (relevance as a category for $\mathbf{z}$). Hence, the cluster assignment probability $p(c \mid \mathbf{z})$ now relies on cluster prototypicality, and we choose a uniform prior $p(c)$ over clusters in $\mathcal{T}$ such that Equation (4) becomes a regularizer encouraging utilization of all clusters.

We now demonstrate that optimizing the ELBO for this hierarchical approach encourages the discovery of prototypical relationships between the latent representations $\mathbf{z}$ and each cluster $c$. Let $\mathcal{X}$ be random variables over $\mathbf{x}$, $\mu_{\mathbf{z}} = \mu_\phi(\mathbf{x})$, and $\sigma_{\mathbf{z}}^2 = \sigma_\phi^2(\mathbf{x})$, we can rewrite Equation (3) as follows:

$$-\mathbb{E}_{q_\phi(c \mid \mathbf{x})}D_{KL}\left(q_\phi(\mathbf{z} \mid \mathbf{x}) \mid\mid p_\theta(\mathbf{z} \mid c)\right) = \mathbb{E}_{q_\phi(\mathbf{z}, c \mid \mathbf{x})}\left[\log p_\theta(\mathbf{z} \mid c)\right] - \mathbb{E}_{q_\phi(\mathbf{z}, c \mid \mathbf{x})}\left[\log q_\phi(\mathbf{z} \mid \mathbf{x})\right]$$

$$\approx \mathbb{E}_{q_\phi(c \mid \mathbf{x})}\left[\int_{\mathbf{z}} \mathcal{N}(\mathbf{z} \mid \mu_{\mathbf{z}}, \sigma_{\mathbf{z}}^2)\log \mathcal{N}(\mathbf{z} \mid \mu_c, \sigma_c^2)d\mathbf{z}\right] + H(\mathcal{Z} \mid \mathcal{X}) \tag{5}$$

The term $H(\mathcal{Z} \mid \mathcal{X})$ can be approximated with Monte Carlo sampling over minibatches. See Appendices A.2 and A.3 for full derivations. The entropy for a multivariate Gaussian with diagonal covariance can be written as: $H(\mathcal{Z} \mid \mathcal{T} = c) = \frac{D}{2}\log(2\pi) + \frac{D}{2} + \frac{1}{2}\sum_d^D \log \sigma_{cd}^2$. As a result, Equation (5) can be expanded as follows:

$$
-\sum_{c\in\mathcal{T}} q_\phi(c|\mathbf{x}) \left[ \frac{D}{2}\log(2\pi) + \frac{1}{2}\sum_d^D \log \sigma_{cd}^2 + \frac{1}{2}\sum_d^D \frac{\sigma_{\mathbf{z}d}^2 + (\mu_{\mathbf{z}d} - \mu_{cd})^2}{\sigma_{cd}^2} \right] + H(\mathcal{Z} \mid \mathcal{X})
$$

$$
\approx -\sum_{c\in\mathcal{T}} q_\phi(c|\mathbf{x}) \left[ \left( H(\mathcal{Z} \mid \mathcal{T} = c) - \frac{D}{2} \right) + \frac{1}{2}\sum_d^D \frac{\sigma_{\mathbf{z}d}^2 + (\mu_{\mathbf{z}d} - \mu_{cd})^2}{\sigma_{cd}^2} \right] + H(\mathcal{Z} \mid \mathcal{X})
$$

$$
\approx \mathbb{E}_{q_\phi(c|\mathbf{x})}[-H(\mathcal{Z} \mid \mathcal{T} = c)] + \mathbb{E}_{q_\phi(c|\mathbf{x})} \left[ \frac{D}{2} - \frac{1}{2}\sum_d^D \frac{\sigma_{\mathbf{z}d}^2 + (\mu_{\mathbf{z}d} - \mu_{cd})^2}{\sigma_{cd}^2} \right] + H(\mathcal{Z} \mid \mathcal{X})
$$

$$
\approx \mathbb{E}_{q_\phi(c|\mathbf{x})}[-H(\mathcal{Z} \mid \mathcal{T} = c)] + H(\mathcal{Z} \mid \mathcal{X}) + G
$$

$$
\approx \mathbb{E}_{p_\theta(c)}[-H(\mathcal{Z} \mid \mathcal{T} = c)] + H(\mathcal{Z} \mid \mathcal{X}) + G \tag{6}
$$

$$
\approx -H(\mathcal{Z} \mid \mathcal{T}) + H(\mathcal{Z} \mid \mathcal{X}) + G
$$

$$
\approx -H(\mathcal{Z} \mid \mathcal{T}) + H(\mathcal{Z}) - H(\mathcal{Z}) + H(\mathcal{Z} \mid \mathcal{X}) + G
$$

$$
\approx I(\mathcal{Z}; \mathcal{T}) - I(\mathcal{Z}; \mathcal{X}) + G \tag{7}
$$

where $G = \mathbb{E}_{q_\phi(c|\mathbf{x})} \left[ \frac{D}{2} - \frac{1}{2}\sum_d^D \frac{\sigma_{\mathbf{z}d}^2 + (\mu_{\mathbf{z}d} - \mu_{cd})^2}{\sigma_{cd}^2} \right]$. The $q_\phi(c \mid \mathbf{x})$ term in Equation (6) can be approximated as $p_\theta(c)$ by the KL divergence term from Equation (4).

As shown in Equation (7), maximizing ELBO maximizes CU. Additionally, the maximization of the negative mutual information term $-I(\mathcal{Z}; \mathcal{X})$ can be understood as optimizing the information bottleneck between the latent representation $\mathbf{z}$ and the input $\mathbf{x}$ [43, 39].

The hierarchical dependency introduced in Section 3.1 embeds abstraction and specificity directly into the prior $p_\theta(\mathbf{z} \mid c)$ via the convex weight $\alpha$. Each prototype's parameters $(\mu_c, \sigma_c^2)$ thus blend broad parental characteristics with fine-grained child traits. Consequently, maximizing the mutual information $I(\mathcal{Z}; \mathcal{T})$ forces $\mathbf{z}$ to distinguish these semantically meaningful hierarchies, rather than discovering arbitrary clusters as would be possible under a flat prior.

### 3.4 Integration of transformation-invariant feature learning

Learning discriminative representations for taxonomic hierarchies from unlabeled real world images is challenging due to high intra- and inter-class variance. For example, ship and bird images may share low-level features (e.g., blue sky, central object) yet belong to distinct semantic categories. To address this, we extend deep taxonomic networks with contrastive learning [3]. The idea is to learn an image representation that is invariant to different transformations such that only the most descriptive features are preserved. Specifically, each image $\mathbf{x}$ is randomly augmented twice, yielding $2N$ views, and we minimize the NT-Xent loss: $\mathcal{L}_{\text{NT-Xent}} = -\log \frac{\exp\big(\text{sim}(\mathbf{h}_i, \mathbf{h}_j)/\tau\big)}{\sum_{k \neq i} \exp\big(\text{sim}(\mathbf{h}_i, \mathbf{h}_k)/\tau\big)}$, where $\mathbf{h}$ is the projection head output on the encoder's features and $\text{sim}$ denotes cosine similarity. The projection head absorbs augmentation variance, letting the encoder focus on invariance. We further introduce cluster-level contrastive learning by projecting the cluster assignment distribution $p(c \mid \mathbf{z})$ and applying the same NT-Xent loss to encourage similar assignments [25, 48, 29]. We refer to Appendix E.1 for the effects on the two loss terms.

## 4 Experiments

**Unsupervised clustering accuracy** Since deep taxonomic networks construct a hierarchy $\mathcal{T}$ without prior knowledge of the true number of classes in the training data, standard unsupervised clustering evaluation methods, such as the Hungarian algorithm [31] which assume a one-to-one mapping between clusters and classes, become unsuitable. Instead, we propose a post-hoc annotation strategy. Given the pre-trained taxonomy $\mathcal{T}$ derived from the training set $\mathcal{D}_{train}$, we first perform a forward pass of $\mathcal{D}_{train}$ through the frozen model to obtain the cluster prototypicality distributions $p(c \mid \mathbf{z}_{train})$ for each training instance $\mathbf{x}_{train}$. Using the ground truth labels $y \in Y$ associated with

$\mathcal{D}_{train}$, we aggregate these distributions for all data points belonging to the same class. This process yields an annotation matrix $\mathcal{A}$ of dimensions $|Y| \times |\mathcal{T}|$. After normalization, each column of $\mathcal{A}$ represents the empirical class distribution $P(Y \mid c)$ for a specific cluster $c \in \mathcal{T}$. Conceptually, clusters higher in the taxonomy (e.g., the root) tend towards a uniform class distribution over a balanced $\mathcal{D}_{train}$, as they represent broader collections of data. Conversely, leaf clusters typically exhibit sharper distributions, indicating a higher concentration of specific classes. To evaluate accuracy on a test dataset $\mathcal{D}_{test}$, we similarly obtain $p(c \mid \mathbf{z}_{test})$ for each test instance $\mathbf{x}_{test}$. The predicted class distributions for $\mathbf{x}_{test}$ is then computed as a weighted sum of the cluster class distributions stored in $\mathcal{A}$, where the weights are given by $p(c \mid \mathbf{z}_{test})$: $\hat{P}(y \mid \mathbf{z}_{test}) = \sum_{c \in \mathcal{T}} p(c \mid \mathbf{z}_{test}) P(Y = y \mid c)$. A key advantage of this evaluation approach is its flexibility. Since no model parameters are updated during this evaluation phase, the deep taxonomic networks can be assessed on datasets with varying sets of classification labels without requiring any retraining or fine-tuning.

**Hierarchical clustering metrics**    In addition to accuracy (ACC) and normalized mutual information (NMI), we also evaluate our approach on hierarchical clustering metrics: leaf purity (LP) and dendrogram purity (DP) [20]. However, our approach differs from other hierarchical clustering methods in that we do not assume the leaf cluster as the final destination of a data point; instead, any cluster can serve as a prototype. We therefore propose probabilistic extensions to both LP and DP. Our probabilistic LP measures cluster homogeneity via soft assignments, and our probabilistic DP computes expected purity for same-class data pairs based on their shared likelihood across all potential clusters, resembling a probabilistic version of lowest common ancestors. Detailed formulations for both metrics are provided in Appendix B.

**Datasets and baselines**    We evaluate the hierarchical clustering performance of deep taxonomic networks on datasets of varying complexities and label sizes: MNIST [23], FashionMNIST (Fashion) [46], CIFAR-10, and CIFAR-100 [21]. For CIFAR-100, we evaluate our approach against both the 20 superclasses (CIFAR-20) and the 100 fine-grained classes. To illustrate the ability to discover taxonomic hierarchies, we additionally train our models on Omniglot [22]. We provide a detailed description of datasets in Appendix C. We compare the hierarchical clustering performance of our approach to deep hierarchical clustering methods such as TreeVAE [29] and DeepECT [30]. We further compare our approach to Cobweb [7], which clusters raw pixels, and to Cobweb+VAE, which uses a VAE model with our encoder and decoder network architectures to produce latent codes for clustering. On CIFAR variants, contrastive learning is applied to the image inputs (Section 3.4), whereas for Cobweb+VAE it is applied only to the latent representations. We additionally use the publicly available code to train TreeVAE on CIFAR-100 with 100 classes using 100 leaf clusters.

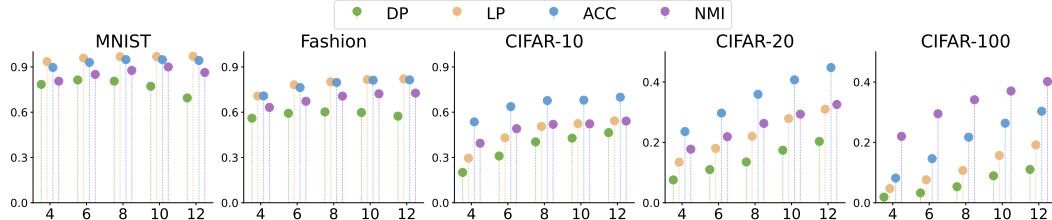

Figure 3: Hierarchical clustering performance on all evaluated datasets at varying depth of $\mathcal{T}$. X-axis: depth, Y-axis: performance.

**Implementation details**    For a direct comparison to baselines, we use the same encoder-decoder architecture from [29] for our approach. Detailed descriptions can be found in Appendix D.1. Our approach initializes the Gaussian parameters of leaf clusters in $\mathcal{T}$ as well as the convex weights $\alpha$ at each branch such that the rest of clusters in the hierarchy can be inferred. To determine the number of clusters in $\mathcal{T}$, we vary tree depth on all evaluated datasets. Figure 3 shows all four hierarchical metrics improve with depth but plateau around depths of 8 to 10 for MNIST and Fashion, and 10 to 12 for CIFAR-10. However, for CIFAR-20 and CIFAR-100, which feature greater class diversity, all metrics consistently rise with increased depth, indicating our approach's scalability with dataset complexity. For a fair comparison across all datasets, we fix a depth of 10—yielding 2047 clusters—for all experiments in this paper. For contrastive learning, we use a two-layer MLP ($512 \rightarrow 64$) with ReLU as the encoder projection head and a single 64-dimensional linear layer for the cluster-level projection on $p(c \mid \mathbf{z})$. We set NT-Xent temperatures to 0.5 (representation) and 0.3 (cluster), with a weighting of 100 to match $\mathcal{L}_{\mathrm{ELBO}}$ following [29]. To stabilize the training of a large $\mathcal{T}$, we introduce two

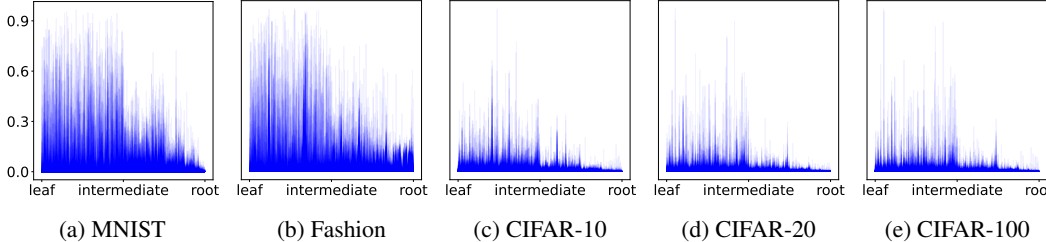

|  (a) MNIST | (b) Fashion | (c) CIFAR-10 | (d) CIFAR-20 | (e) CIFAR-100 |

Figure 4: Prototypicality $p(c \mid \mathbf{z})$ on test data over $\mathcal{T}$. X-axis: Cluster indices of a flattened complete binary tree, ordered left-to-right starting with its $2^{10}$ leaf clusters. Y-axis: $p(c \mid \mathbf{z})$.

additional entropy regularization terms that penalize biased higher-level parent clusters (i.e., $\alpha \approx 1$) and indistinguishable lower-level clusters where their KL divergences are close. See Appendix D.2 for additional details. We train deep taxonomic networks for 400 epochs on all datasets with Adam [17] at a constant learning rate of $1 \times 10^{-3}$ and a batch size of 256.

## 5  Results and discussions

### 5.1  Prototypicality across the learned taxonomy

Figure 4 shows the prototypicality $p(c \mid \mathbf{z})$ for unlabeled test samples across evaluated datasets. We observe that the most prototypical cluster can occur at any depth in the taxonomy, but is predominantly found in lower-level clusters. This pattern indicates that finer clusters capture the most informative features of each data point [4], whereas higher-level clusters such as root, which reflect more generalized averages, rarely serve as prototypes. Notably, MNIST and Fashion exhibit more intermediate-level prototypes—likely because leaf clusters become too specific (e.g., unique handwriting styles) to represent a general prototype (Figure 5d). By contrast, the greater complexity and variance in CIFAR-10, CIFAR-20, and CIFAR-100 necessitates deeper hierarchies to reach a similar level of feature specificity.

Table 1: Hierarchical clustering performance (%) with standard deviations on 4 evaluated datasets. [†]: Results are adopted from [29]. [*]: Contrastive learning is applied during training. Results are averaged over 10 random seeds.

| Dataset | Models | DP | LP | ACC | NMI |
|---|---|---|---|---|---|
| MNIST | Cobweb | $77.4_{\pm 1.1}$ | $90.7_{\pm 0.8}$ | $88.2_{\pm 2.1}$ | $78.1_{\pm 3.4}$ |
| | Cobweb + VAE | $72.6_{\pm 2.1}$ | $87.8_{\pm 0.9}$ | $89.3_{\pm 1.4}$ | $78.6_{\pm 2.9}$ |
| | DeepECT[†] | $74.6_{\pm 5.9}$ | $90.7_{\pm 3.2}$ | $74.9_{\pm 6.2}$ | $76.7_{\pm 4.2}$ |
| | TreeVAE[†] | $\mathbf{87.9}_{\pm 4.9}$ | $96.0_{\pm 1.9}$ | $90.2_{\pm 7.5}$ | $\mathbf{90.0}_{\pm 4.6}$ |
| | **DeepTaxonNet** | $76.6_{\pm 2.3}$ | $\mathbf{96.7}_{\pm 0.2}$ | $\mathbf{94.8}_{\pm 0.2}$ | $88.1_{\pm 0.6}$ |
| Fashion | Cobweb | $57.2_{\pm 1.7}$ | $78.5_{\pm 0.8}$ | $75.2_{\pm 1.7}$ | $66.7_{\pm 2.1}$ |
| | Cobweb + VAE | $56.6_{\pm 0.8}$ | $72.1_{\pm 2.0}$ | $75.1_{\pm 1.7}$ | $66.5_{\pm 2.9}$ |
| | DeepECT[†] | $44.9_{\pm 3.3}$ | $67.8_{\pm 1.4}$ | $51.8_{\pm 5.7}$ | $57.7_{\pm 3.7}$ |
| | TreeVAE[†] | $54.4_{\pm 2.4}$ | $71.4_{\pm 2.0}$ | $63.6_{\pm 3.3}$ | $64.7_{\pm 1.4}$ |
| | **DeepTaxonNet** | $\mathbf{59.8}_{\pm 0.8}$ | $\mathbf{81.6}_{\pm 0.3}$ | $\mathbf{81.2}_{\pm 0.2}$ | $\mathbf{72.2}_{\pm 0.2}$ |
| CIFAR-10* | Cobweb + VAE | $10.02_{\pm 0.41}$ | $18.91_{\pm 0.27}$ | $16.36_{\pm 0.34}$ | $2.50_{\pm 0.57}$ |
| | DeepECT[†] | $10.01_{\pm 0.02}$ | $10.30_{\pm 0.40}$ | $10.31_{\pm 0.39}$ | $0.18_{\pm 0.10}$ |
| | TreeVAE[†] | $35.30_{\pm 1.15}$ | $53.85_{\pm 1.23}$ | $52.98_{\pm 1.34}$ | $41.44_{\pm 1.13}$ |
| | **DeepTaxonNet** | $\mathbf{42.89}_{\pm 1.12}$ | $\mathbf{54.31}_{\pm 0.63}$ | $\mathbf{67.97}_{\pm 0.91}$ | $\mathbf{51.83}_{\pm 0.70}$ |
| CIFAR-20* | Cobweb + VAE | $5.01_{\pm 0.16}$ | $10.94_{\pm 0.09}$ | $9.39_{\pm 0.40}$ | $3.30_{\pm 0.63}$ |
| | DeepECT[†] | $5.28_{\pm 0.18}$ | $6.97_{\pm 0.69}$ | $6.97_{\pm 0.69}$ | $1.71_{\pm 0.86}$ |
| | TreeVAE[†] | $10.44_{\pm 0.38}$ | $24.16_{\pm 0.65}$ | $21.82_{\pm 0.77}$ | $17.80_{\pm 0.42}$ |
| | **DeepTaxonNet** | $\mathbf{17.40}_{\pm 0.23}$ | $\mathbf{27.87}_{\pm 0.47}$ | $\mathbf{40.72}_{\pm 0.39}$ | $\mathbf{29.36}_{\pm 0.47}$ |

## 5.2 Hierarchical clustering performance

We evaluate deep taxonomic networks against established baselines on four datasets (Table 1). Overall, deep taxonomic networks outperform all baselines in both hierarchical clustering accuracy (ACC, NMI) and hierarchical purity (DP, LP) with the exception on MNIST dataset. We attribute MNIST's lower DP and NMI to its low inter- and intra-class variance, which causes bottom-level clusters to capture handwriting idiosyncrasies rather than digit-level features (Figure 5d). By contrast, on Fashion our approach outperforms all baselines by a large margin. When jointly trained with contrastive learning on CIFAR-10/20, our approach outperforms baselines by learning more consistent hierarchies and achieving higher classification accuracy. Notably, Cobweb models trained on raw pixels perform competitively on MNIST and Fashion due to its feature-independence assumption, which can be effective for simpler images where individual pixels alone may suffice to characterize the features [32, 24]. However, Cobweb gains little from VAE embeddings, where it inherits feature dependencies from the encoder networks. In contrast, our approach—despite assuming diagonal covariance—jointly optimizes encoding and clustering end-to-end, implicitly learning feature dependencies and achieving superior performance on both simple and complex, real-world images.

We argue that our approaches benefit from the learned intermediate prototypes. Leaf purity (LP) measures the class entropy at the leaves (the nodes with the most fine-grained semantic classes). Table 1 shows that despite having substantially more leaf nodes ($2^{10}$ nodes) than TreeVAE and DeepECT (10 nodes, corresponding to 10 classes), our approach's leaf purity is comparable to the baseline approaches (it has higher LP), suggesting robust, consistent, and well-structured fine-grained semantic classes at the leaf level. In other words, while other approaches have 10 leaves corresponding to the 10 classes, ours can identify more subclasses that are inherent in the data, but are not explicitly called out in the labeling. This is beneficial for classification because our approach enables the model to better disentangle subtle semantic differences that might found similar across labels (e.g., a digit '4' that is similar to a '9' in MNIST, or an 'automobile' that is similar to a 'truck' in CIFAR-10).

An interesting observation from the leaf-only approaches (TreeVAE and DeepECT) is that ACC is lower than LP. This is expected as leaf-only approaches assume the same number of leaves as the classes so the classification accuracy depends on both the quality of leaf nodes representing a class (LP) and the routing quality that successfully brings data to a leaf node, and hence ACC should be upper bounded by LP. However, we observe a substantial increase in ACC in our approach (e.g., on CIFARs) compared to TreeVAE, despite having a similar LP. Notably, ACC is not upper-bounded by LP, as is the case with the leaf-only approach. This result indicates that the additional ACC gain in our approach is a benefit of utilizing additional intermediate nodes. In other words, by utilizing all levels, our approach can correctly capture in intermediate nodes what would otherwise be misclassified in the leaf nodes of leaf-only approaches, results in better classification accuracy.

**Adaptation to new classification task without re-training** Our approach enables flexible classification across different label granularities without the need for re-training. Specifically, we used the pre-trained, frozen hierarchy from CIFAR-20 in Table 1 to evaluate the 100 fine-grained classes using the evaluation method described in Section 4. We find that deep taxonomic networks outperform TreeVAE, which is re-trained by growing up to 100 cluster nodes, on all metrics, with accuracy of $26.36_{\pm 0.36}$ compare to $11.98_{\pm 0.18}$. This result suggests that our approach is able to adapt to different classification objectives by utilizing the rich hierarchical prototype clusters.

Table 2: CIFAR-100 hierarchical classification results (%) on TreeVAE and deep taxonomic network. [†]: The same, frozen model used in Table 1 on CIFAR-20. [*]: Contrastive learning is applied during training TreeVAE. Results are averaged over 10 random seeds.

| Dataset | Models | DP | LP | ACC | NMI |
|---------|--------|-----|-----|------|------|
| CIFAR-100* | TreeVAE | $3.77_{\pm 0.08}$ | $12.11_{\pm 0.11}$ | $11.98_{\pm 0.18}$ | $27.57_{\pm 0.20}$ |
| | **DeepTaxonNet**[†] | $\mathbf{8.29}_{\pm 0.26}$ | $\mathbf{15.68}_{\pm 0.38}$ | $\mathbf{26.36}_{\pm 0.36}$ | $\mathbf{37.03}_{\pm 0.33}$ |

**Hierarchical classification on pre-trained features** While we adopt the same encoder-decoder model architecture as TreeVAE for a direct comparison, we argue that the performance of our

approach benefit from models that learn a stronger feature representation. Inspired by L2H [34], we perform hierarchical clustering on top of unsupervised pre-trained image features from DINOv2 [33].

Specifically, we replace the encoder and decoder with one linear layer each. The encoder maps the DINOv2 feature to a 128 dimensional hidden representation, and the decoder maps it back to the original feature dimension. We additionally disabled contrastive learning in this setting. Our goal is to assess whether our methods stay robust with a stronger representation while remaining directly comparable to the L2H baseline, which does not use contrastive learning.

Table 3 shows that when using stronger image features, our approaches match the performance from L2H in flat clustering metrics (ACC, NMI) on all evaluated datasets. On hierarchical metrics (DP, LP), our approaches achieve higher LP

Table 3: Hierarchical classification results (%) on CIFAR-10 and CIFAR-100 using DINOv2 features. *: Results are adopted from [34]. Underscore denotes the second best result.

| Dataset | Models | DP | LP | ACC | NMI |
|---|---|---|---|---|---|
| CIFAR-10 | L2H-TEMI* | 90.2 | 95.8 | 95.6 | 90.1 |
| | L2H-Turtle* | **98.8** | 99.5 | **99.5** | **98.5** |
| | **DeepTaxonNet** | 88.0 | **99.6** | 99.1 | 97.4 |
| CIFAR-100 | L2H-TEMI* | 50.2 | 69.8 | 68.2 | 77.8 |
| | L2H-Turtle* | **80.3** | 89.6 | **89.6** | **91.7** |
| | **DeepTaxonNet** | 71.0 | **93.0** | 88.7 | 89.4 |

despite having much deeper hierarchical clusters and more leaf nodes. While our approaches reach a lower, but comparable DP than L2H variants, we argue that it is because our approaches learn about $100\times$ more intermediate nodes. These results suggest that the performance of our approach is not limited to the current model architecture choice, and can benefit from a stronger feature representation, while additionally offering much deeper hierarchical clusters, including intermediate clusters, and without requiring prior knowledge of the number of labels.

## 5.3 Discovery of hierarchical prototypes

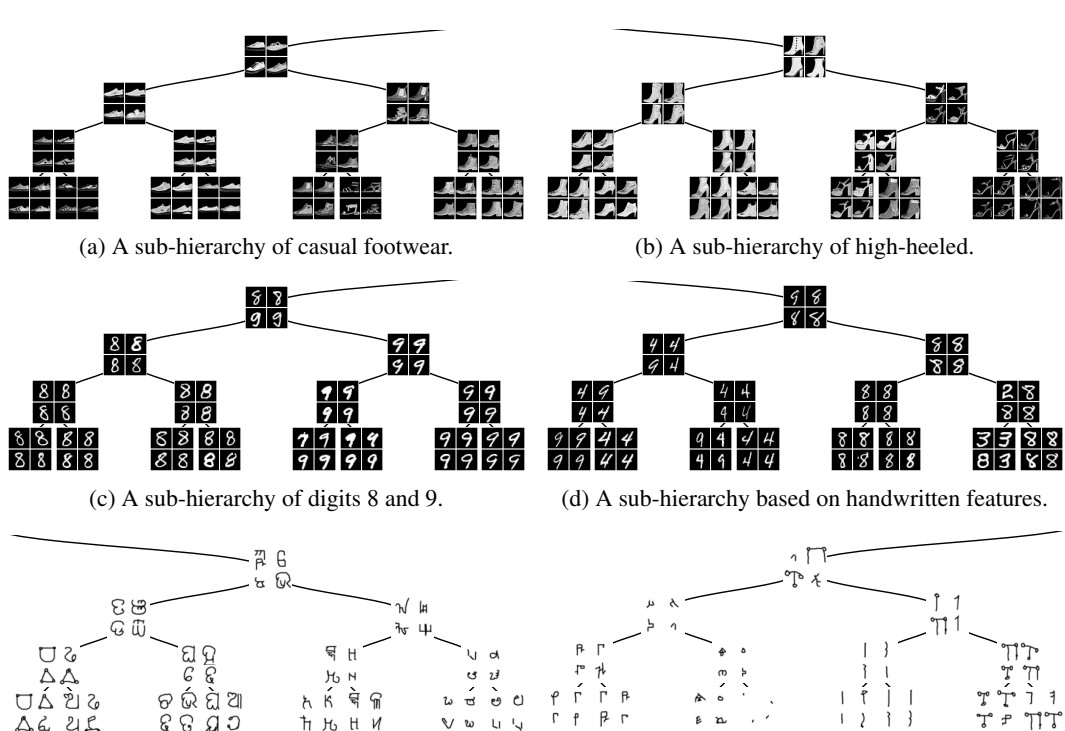

(a) A sub-hierarchy of casual footwear.

(b) A sub-hierarchy of high-heeled.

(c) A sub-hierarchy of digits 8 and 9.

(d) A sub-hierarchy based on handwritten features.

(e) A sub-hierarchy of rounded and angular characters.

(f) A sub-hierarchy of characters with dots and lines.

Figure 5: Examples of sub-hierarchy discovered by deep taxonomic networks on MNIST (5c, 5d), Fashion (5a, 5b) and Omniglot (5e, 5f). Images are sampled from the test set per cluster by likelihood.

For the qualitative results, we present examples of sub-hierarchies discovered by deep taxonomic network models trained on CIFAR-10 (Figures 1a and 1b), MNIST (Figures 5c and 5d), Fashion (Figures 5a and 5b), and Omniglot (Figures 5e and 5f).

On CIFAR-10, our approach uncovers interpretable hierarchies of both animal and vehicle classes. Figure 1a splits far-away deer-like silhouettes from close-up horse shots, then refines by silhouette vs. natural-color context and grazing vs. ridden scenes. Figure 1b divides service vehicles (industrial machinery vs. emergency fleets) from passenger cars, then partitions compact models vs. red, white and blue hatchbacks by form and hue.

Our approach also constructs hierarchies reflecting established categories in Fashion. Figure 5a organizes casual footwear such as sneakers and flat shoes, while Figure 5b distinguishes styles of high-heeled shoes. On MNIST, our model similarly forms hierarchies based on visual criteria. Figure 5c partitions digits clearly by class, separating clusters of "8"s from "9"s. Figure 5d captures finer similarities in handwritten styles, grouping visually similar "4"s with "9"s, and certain "3"s with "8"s, highlighting the model's ability to learn perceptually relevant features beyond labels.

Deep taxonomic networks similarly discover coherent structures from diverse handwritten characters across numerous alphabets in Omniglot. Figure 5e groups characters by fundamental visual traits, separating predominantly rounded, continuous strokes from more angular, geometric features. Figure 5f further distinguishes characters by their elemental composition and structure, grouping line-based shapes (e.g., $\rho$- or $\Gamma$-like) apart from those with dots or fragmented strokes, and differentiating simple vertical strokes ($|$) from structures like $\top$ or $\Pi$. These examples highlight our approach's ability to learn and organize abstract structural properties within complex visual data.

Overall, these qualitative examples show that deep taxonomic networks are able to discover rich, interpretable hierarchical prototypes, capturing both coarse-grained semantic categories and fine-grained visual distinctions within the data.

## 6 Conclusion, limitations, and future work

In this paper, we propose deep taxonomic networks, a novel deep latent variable approach with a complete binary tree mixture-of-Gaussians prior that learns a taxonomic hierarchy over unlabeled image data by finding the most prototypical clusters. Contrary to previous hierarchical clustering methods, deep taxonomic networks do not reply on the true label size to construct the hierarchy, and treat every cluster as the potential prototype of a datum. We analytically show that optimizing the learning objective of deep taxonomic networks maximizes the ability to discover hierarchical prototypes of the data. Our empirical results show that our approach outperforms baseline hierarchical clustering methods on datasets of varying complexity and with varying label sizes by a large margin. This is achieved through our novel evaluation method that leverages prototype clusters discovered at all hierarchical levels, and that can use the learned hierarchy to support a new classification objectives on the fly. Finally, we present qualitative results that show examples of subsets of discovered taxonomic hierarchies learned from various datasets, where the hierarchies contain interpretable hierarchical prototypes. Our findings suggest that deep taxonomic networks are a powerful new unsupervised hierarchical clustering approach, with the potential to form human-like concepts.

**Limitations and future work**    While the pre-allocated (up to a compute constraint) complete binary tree prior makes the analysis straightforward, this assumption introduces an inductive bias that a dataset should have balanced feature splits. However, this assumption might lead to a degraded taxonomic hierarchy if the dataset is dominated by unbalanced data with low inter-class feature entropy. Future work should focus on developing a dynamic mixture-of-Gaussians prior that adapts its structure to the dataset. In addition, while the scope of this work is to study the discovery of taxonomic hierarchy within VAE-based framework, future work should explore the generative capabilities of deep taxonomic networks to produce high quality data at multiple levels of granularity.

## Acknowledgments

We would like to thank Douglas Fisher, Kyle Moore, Jesse Roberts, Pat Langley, Nicki Barari, Xin Lian, Zsolt Kira, and Ziqiao Ma for discussions.

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

# A   Additional Derivations

## A.1   Full EBLO Derivation

We begin from

$$
\begin{aligned}
\log p_\theta(\mathbf{x}) &= \log \int_{\mathbf{z}} \sum_{c \in \mathcal{T}} p_\theta(\mathbf{x} \mid \mathbf{z}) \, p_\theta(\mathbf{z} \mid c) \, p_\theta(c) \; d\mathbf{z} \\
&= \log \int_{\mathbf{z}} \sum_{c \in \mathcal{T}} q_\phi(\mathbf{z}, c \mid \mathbf{x}) \, \frac{p_\theta(\mathbf{x} \mid \mathbf{z}) \, p_\theta(\mathbf{z} \mid c) \, p_\theta(c)}{q_\phi(\mathbf{z}, c \mid \mathbf{x})} \; d\mathbf{z} \\
&= \log \mathbb{E}_{q_\phi(\mathbf{z},c\mid\mathbf{x})} \left[ \frac{p_\theta(\mathbf{x} \mid \mathbf{z}) \, p_\theta(\mathbf{z} \mid c) \, p_\theta(c)}{q_\phi(\mathbf{z}, c \mid \mathbf{x})} \right] \geq \mathbb{E}_{q_\phi(\mathbf{z},c\mid\mathbf{x})} \left[ \log \frac{p_\theta(\mathbf{x} \mid \mathbf{z}) \, p_\theta(\mathbf{z} \mid c) \, p_\theta(c)}{q_\phi(\mathbf{z}, c \mid \mathbf{x})} \right]
\end{aligned}
\tag{8}
$$

We now expand the expectation in Equation (8):

$$
\begin{aligned}
\mathcal{L}_{ELBO}(\phi, \theta) = \; & \mathbb{E}_{q_\phi(\mathbf{z},c\mid\mathbf{x})} \Big[ \log p_\theta(\mathbf{x} \mid \mathbf{z}) \Big] + \mathbb{E}_{q_\phi(\mathbf{z},c\mid\mathbf{x})} \Big[ \log p_\theta(\mathbf{z} \mid c) \Big] \\
& + \mathbb{E}_{q_\phi(\mathbf{z},c\mid\mathbf{x})} \Big[ \log p_\theta(c) \Big] - \mathbb{E}_{q_\phi(\mathbf{z},c\mid\mathbf{x})} \Big[ \log q_\phi(\mathbf{z}, c \mid \mathbf{x}) \Big].
\end{aligned}
\tag{9}
$$

Use the factorization $q_\phi(\mathbf{z}, c \mid \mathbf{x}) = q_\phi(c \mid \mathbf{x}) \, q_\phi(\mathbf{z} \mid \mathbf{x})$ to split the last term:

$$
\mathbb{E}_{q_\phi(\mathbf{z},c\mid\mathbf{x})} \Big[ \log q_\phi(\mathbf{z}, c \mid \mathbf{x}) \Big] = \mathbb{E}_{q_\phi(c\mid\mathbf{x})} \mathbb{E}_{q_\phi(\mathbf{z}\mid\mathbf{x})} \Big[ \log q_\phi(\mathbf{z} \mid \mathbf{x}) \Big] + \mathbb{E}_{q_\phi(c\mid\mathbf{x})} \Big[ \log q_\phi(c \mid \mathbf{x}) \Big].
\tag{10}
$$

Plugging Equation (10) into Equation (9), regroup terms:

1. **Reconstruction term**

$$
\mathbb{E}_{q_\phi(\mathbf{z}\mid\mathbf{x})} \Big[ \log p_\theta(\mathbf{x} \mid \mathbf{z}) \Big] \quad \text{(matches Equation (2))}
$$

2. **Cluster-conditioned KL**

$$
\mathbb{E}_{q_\phi(c\mid\mathbf{x})} \Big[ \mathbb{E}_{q_\phi(\mathbf{z}\mid\mathbf{x})} \Big[ \log p_\theta(\mathbf{z} \mid c) - \log q_\phi(\mathbf{z} \mid \mathbf{x}) \Big] \Big] = -\mathbb{E}_{q_\phi(c\mid\mathbf{x})} \Big[ D_{KL}(q_\phi(\mathbf{z} \mid \mathbf{x}) \,\|\, p_\theta(\mathbf{z} \mid c)) \Big]
$$

matches Equation (3).

3. **Cluster-prior KL**

$$
\mathbb{E}_{q_\phi(c\mid\mathbf{x})} \Big[ \log p_\theta(c) - \log q_\phi(c \mid \mathbf{x}) \Big] = -D_{KL}\big(q_\phi(c \mid \mathbf{x}) \,\|\, p_\theta(c)\big) \quad \text{(matches Equation (4))}.
$$

Putting it all together,

$$
\mathcal{L}_{ELBO}(\phi, \theta) = \underbrace{\mathbb{E}_{q_\phi(\mathbf{z}\mid\mathbf{x})} \Big[ \log p_\theta(\mathbf{x} \mid \mathbf{z}) \Big]}_{Equation~(2)} - \underbrace{\mathbb{E}_{q_\phi(c\mid\mathbf{x})} \Big[ D_{KL}\big(q_\phi(\mathbf{z} \mid \mathbf{x}) \,\|\, p_\theta(\mathbf{z} \mid c)\big) \Big]}_{Equation~(3)} - \underbrace{D_{KL}\big(q_\phi(c \mid \mathbf{x}) \,\|\, p_\theta(c)\big)}_{Equation~(4)}.
$$

## A.2 Derivation of the First Term in Equation (5)

$$\mathbb{E}_{q_\phi(\mathbf{z},c|\mathbf{x})}[\log p_\theta(\mathbf{z} \mid c)]$$

$$= \sum_{c\in\mathcal{T}} \int_{\mathbf{z}} q_\phi(\mathbf{z}, c \mid \mathbf{x}) \, \log p_\theta(\mathbf{z} \mid c) \, d\mathbf{z}$$

$$= \sum_{c\in\mathcal{T}} \int_{\mathbf{z}} q_\phi(c \mid \mathbf{x}) \, q_\phi(\mathbf{z} \mid \mathbf{x}) \, \log p_\theta(\mathbf{z} \mid c) \, d\mathbf{z}$$

$$= \sum_{c\in\mathcal{T}} q_\phi(c \mid \mathbf{x}) \int_{\mathbf{z}} \mathcal{N}(\mathbf{z} \mid \mu_{\mathbf{z}}, \sigma_{\mathbf{z}}^2) \, \log \mathcal{N}(\mathbf{z} \mid \mu_c, \sigma_c^2) \, d\mathbf{z}$$

$$= \sum_{c\in\mathcal{T}} q_\phi(c \mid \mathbf{x}) \int \mathcal{N}(\mathbf{z} \mid \mu_{\mathbf{z}}, \sigma_{\mathbf{z}}^2) \, \log \mathcal{N}(\mathbf{z} \mid \mu_c, \sigma_c^2) \, d\mathbf{z}$$

$$= \sum_{c} q_\phi(c \mid \mathbf{x}) \int \mathcal{N}(\mathbf{z} \mid \mu_{\mathbf{z}}, \sigma_{\mathbf{z}}^2) \left[ -\tfrac{D}{2}\log(2\pi) - \tfrac{1}{2}\log\det\Sigma_c - \tfrac{1}{2}(\mathbf{z}-\mu_c)^\top\Sigma_c^{-1}(\mathbf{z}-\mu_c) \right] d\mathbf{z}$$

$$= \sum_{c} q_\phi(c \mid \mathbf{x}) \left\{ -\tfrac{D}{2}\log(2\pi) - \tfrac{1}{2}\log\det\Sigma_c - \tfrac{1}{2}\underbrace{\mathbb{E}_{\mathcal{N}(\mathbf{z}|\mu_{\mathbf{z}},\sigma_{\mathbf{z}}^2)}\big[(\mathbf{z}-\mu_c)^\top\Sigma_c^{-1}(\mathbf{z}-\mu_c)\big]}_{\mathrm{tr}(\Sigma\Sigma_c^{-1})+(\mu_{\mathbf{z}}-\mu_c)^\top\Sigma_c^{-1}(\mu_{\mathbf{z}}-\mu_c)} \right\}$$

$$= -\sum_{c} q_\phi(c \mid \mathbf{x}) \left[ \frac{D}{2}\log(2\pi) + \frac{1}{2}\log\det\Sigma_c + \frac{1}{2}\left( \mathrm{tr}(\Sigma\Sigma_c^{-1}) + (\mu_{\mathbf{z}}-\mu_c)^\top\Sigma_c^{-1}(\mu_{\mathbf{z}}-\mu_c) \right) \right]$$

For diagonal covariances $\Sigma = \mathrm{diag}(\sigma_{\mathbf{z}d}^2)$, $\Sigma_c = \mathrm{diag}(\sigma_{cd}^2)$, one has

$$\log\det\Sigma_c = \sum_{d=1}^{D}\log\sigma_{cd}^2, \quad \mathrm{tr}(\Sigma\Sigma_c^{-1}) = \sum_{d=1}^{D}\frac{\sigma_{\mathbf{z}d}^2}{\sigma_{cd}^2}, \quad (\mu_{\mathbf{z}}-\mu_c)^\top\Sigma_c^{-1}(\mu_{\mathbf{z}}-\mu_c) = \sum_{d=1}^{D}\frac{(\mu_{\mathbf{z}d}-\mu_{cd})^2}{\sigma_{cd}^2}.$$

Hence the final result:

$$= -\sum_{c} q_\phi(c \mid \mathbf{x}) \left[ \frac{D}{2}\log(2\pi) + \frac{1}{2}\sum_{d=1}^{D}\log\sigma_{cd}^2 + \frac{1}{2}\sum_{d=1}^{D}\frac{\sigma_{\mathbf{z}d}^2}{\sigma_{cd}^2} + \frac{1}{2}\sum_{d=1}^{D}\frac{(\mu_{\mathbf{z}d}-\mu_{cd})^2}{\sigma_{cd}^2} \right] \tag{11}$$

$$= -\sum_{c\in\mathcal{T}} q_\phi(c|\mathbf{x}) \left[ \frac{D}{2}\log(2\pi) + \frac{1}{2}\sum_{d}\log\sigma_{cd}^2 + \frac{1}{2}\sum_{d}\frac{\sigma_{\mathbf{z}d}^2 + (\mu_{\mathbf{z}d}-\mu_{cd})^2}{\sigma_{cd}^2} \right] \tag{12}$$

## A.3 Derivation of the Second Term in Equation (5)

By definition of the conditional (differential) entropy under the variational posterior,

$$H(\mathcal{Z} \mid \mathcal{X}) = -\mathbb{E}_{p(\mathbf{x})}\Big[\mathbb{E}_{q_\phi(\mathbf{z},c|\mathbf{x})}\big[\log q_\phi(\mathbf{z} \mid \mathbf{x})\big]\Big].$$

Because the posterior factorizes as

$$q_\phi(\mathbf{z}, c \mid \mathbf{x}) = q_\phi(c \mid \mathbf{x}) \, q_\phi(\mathbf{z} \mid \mathbf{x}),$$

and $q_\phi(c \mid \mathbf{x})$ does not depend on $z$, the inner expectation simplifies to

$$\mathbb{E}_{q_\phi(\mathbf{z},c|\mathbf{x})}\big[\log q_\phi(\mathbf{z} \mid \mathbf{x})\big] = \mathbb{E}_{q_\phi(\mathbf{z}|\mathbf{x})}\big[\log q_\phi(\mathbf{z} \mid \mathbf{x})\big].$$

Hence,

$$H(\mathcal{Z} \mid \mathcal{X}) = -\mathbb{E}_{p(\mathbf{x})}\Big[\mathbb{E}_{q_\phi(\mathbf{z}|\mathbf{x})}\big[\log q_\phi(\mathbf{z} \mid \mathbf{x})\big]\Big].$$

We approximate the outer expectation over $p(\mathbf{x})$ by an empirical average over $N$ training samples $\{\mathbf{x}^{(n)}\}_{n=1}^{N}$, and the inner expectation over $q_\phi(\mathbf{z} \mid \mathbf{x}^{(n)})$ by $M$ Monte Carlo samples $\{\mathbf{z}^{(n,m)}\}_{m=1}^{M}$. Thus:

$$H(\mathcal{Z} \mid \mathcal{X}) \approx -\frac{1}{N} \sum_{n=1}^{N} \underbrace{\mathbb{E}_{q_\phi(\mathbf{z}\mid\mathbf{x}^{(n)})}\big[\log q_\phi(\mathbf{z} \mid \mathbf{x}^{(n)})\big]}_{\text{approx. by } M \text{ samples}}$$

$$\approx -\frac{1}{N} \sum_{n=1}^{N} \frac{1}{M} \sum_{m=1}^{M} \log q_\phi\big(\mathbf{z}^{(n,m)} \mid \mathbf{x}^{(n)}\big).$$

Equivalently:

$$H(\mathcal{Z} \mid \mathcal{X}) \approx -\frac{1}{NM} \sum_{n=1}^{N} \sum_{m=1}^{M} \log q_\phi\big(\mathbf{z}^{(n,m)} \mid \mathbf{x}^{(n)}\big),$$

where
$$\mathbf{x}^{(n)} \sim \text{training data}, \qquad \mathbf{z}^{(n,m)} \sim q_\phi\big(\mathbf{z} \mid \mathbf{x}^{(n)}\big) \quad \text{via reparameterization trick [18]}.$$

# B  Probabilistic Hierarchical Clustering Metrics

## B.1  Probabilistic Dendrogram Purity

We propose a probabilistic extension to Dendrogram Purity (DP) to suit our model where any cluster $c$ can serve as a prototype and data points $\mathbf{x}$ (with representations $\mathbf{z}$) have soft assignments $p(c|\mathbf{z})$ to all clusters in the hierarchy. Traditional DP relies on the purity of subtrees at the Lowest Common Ancestor (LCA) for pairs of same-class data points. In our probabilistic DP ($DP_{prob}$), for any two data points $\mathbf{x}_i$ and $\mathbf{x}_j$ belonging to the same ground-truth class $G_k$, we first define a shared cluster likelihood for each cluster $c$ as $S_c(\mathbf{x}_i, \mathbf{x}_j)$ (Equation 13). The contribution of this pair to $DP_{prob}$ is then the expected purity over all clusters $c$, where the purity of an individual cluster $P(c, G_k)$ (as defined in Equation 14) is weighted by the normalized likelihood $S_c(\mathbf{x}_i, \mathbf{x}_j)$ that $c$ is a shared cluster for the pair (see Equation 15). The final $DP_{prob}$ is the average of these expected purities across all same-class data pairs (as defined in Equation 16).

The mathematical formulation is as follows: Let $\mathbf{x}$ denote a data point and $\mathbf{z}$ its corresponding representation. Let $c$ be an arbitrary cluster (node) within the hierarchical structure $\mathcal{T}$. The probabilistic assignment of data point $\mathbf{x}$ to cluster $c$ is given by $p(c|\mathbf{z})$. Let $G_k$ denote the $k$-th ground-truth class.

The shared cluster likelihood for any cluster $c \in \mathcal{T}$ for a pair of data points $(\mathbf{x}_i, \mathbf{x}_j)$ is defined as:
$$S_c(\mathbf{x}_i, \mathbf{x}_j) = p(c|\mathbf{z}_i)p(c|\mathbf{z}_j) \tag{13}$$

The probabilistic purity of an individual cluster $c \in \mathcal{T}$ with respect to a ground-truth class $G_k$ is:
$$P(c, G_k) = \frac{\sum_{\mathbf{x}_l \in G_k} p(c|\mathbf{z}_l)}{\sum_{\text{all } \mathbf{x}_m} p(c|\mathbf{z}_m)} \tag{14}$$

For a pair of data points $(\mathbf{x}_i, \mathbf{x}_j)$ that both belong to the same ground-truth class $G_k$, their contribution to the Dendrogram Purity, termed the expected purity $E_P(\mathbf{x}_i, \mathbf{x}_j, G_k)$, is calculated as:
$$E_P(\mathbf{x}_i, \mathbf{x}_j, G_k) = \frac{\sum_{c \in \mathcal{T}} (S_c(\mathbf{x}_i, \mathbf{x}_j) \times P(c, G_k))}{\sum_{c' \in \mathcal{T}} S_{c'}(\mathbf{x}_i, \mathbf{x}_j)} \tag{15}$$

The overall probabilistic Dendrogram Purity ($DP_{prob}$) is then the average of these expected purities over all distinct pairs of data points belonging to the same ground-truth class:
$$DP_{prob} = \frac{1}{Z} \sum_k \sum_{\substack{\mathbf{x}_i, \mathbf{x}_j \in G_k \\ i \neq j}} E_P(\mathbf{x}_i, \mathbf{x}_j, G_k) \tag{16}$$

where $Z$ is the total number of such distinct pairs, calculated as $Z = \sum_k \binom{|G_k|}{2}$, and $E_P(\mathbf{x}_i, \mathbf{x}_j, G_k)$ is the expected purity for the pair $(\mathbf{x}_i, \mathbf{x}_j)$ from class $G_k$.

## B.2 Probabilistic Leaf Purity

Standard Leaf Purity (LP) evaluates the homogeneity of leaf clusters in a hierarchy with respect to ground-truth classes. It typically measures the proportion of data points in leaf clusters that belong to the majority class within each respective leaf. To adapt this metric for our model, where data points $\mathbf{x}$ (with representations $\mathbf{z}$) have soft assignments $p(c|\mathbf{z})$ to clusters $c$ in the hierarchy, we define Probabilistic Leaf Purity ($LP_{prob}$). This formulation specifically considers the leaf clusters $\mathbf{L}$ of the hierarchy and utilizes the probabilistic assignments $p(L|\mathbf{z})$ for $L \in \mathbf{L}$ as fractional counts of data point membership.

The mathematical formulation is as follows: Let $\mathbf{L}$ be the set of all leaf clusters in the hierarchy. Let $G_k$ denote the $k$-th ground-truth class. The probabilistic assignment of data point $\mathbf{x}$ (with representation $\mathbf{z}$) to a specific leaf cluster $L \in \mathbf{L}$ is given by $p(L|\mathbf{z})$.

For each leaf cluster $L \in \mathcal{L}$, we first determine the total probabilistic mass contributed by each ground-truth class $G_k$:

$$M(L, G_k) = \sum_{\mathbf{x}_i \in G_k} p(L|\mathbf{z}_i) \tag{17}$$

The majority ground-truth class for leaf cluster $L$, denoted $G_L^*$, is the class that maximizes this probabilistic mass:

$$G_L^* = \arg\max_{G_k} M(L, G_k) \tag{18}$$

The probabilistic mass of correctly assigned data points within leaf cluster $L$ is therefore $M(L, G_L^*)$.

The overall Probabilistic Leaf Purity ($LP_{prob}$) is then calculated as the ratio of the sum of these correctly assigned probabilistic masses across all leaf clusters to the sum of all probabilistic masses assigned to any leaf cluster by any data point:

$$LP_{prob} = \frac{\sum_{L \in \mathbf{L}} M(L, G_L^*)}{\sum_{L' \in \mathbf{L}} \sum_{\text{all } \mathbf{x}_j} p(L'|\mathbf{z}_j)} \tag{19}$$

The denominator represents the total probabilistic assignment of all data points to the set of leaf clusters. If, for every data point $\mathbf{x}_j$, the sum of its probabilities to leaf clusters $\sum_{L' \in \mathcal{L}} p(L'|\mathbf{z}_j)$ equals 1 (meaning each point's probability mass for leaf assignment is fully accounted for among the leaves), the denominator simplifies to $N$, the total number of data points.

# C  Datasets

For our experimental evaluation, we utilize several standard image datasets with varying characteristics and complexity. These datasets were chosen to evaluate our model's performance across different image types, resolutions, and numbers of classes and samples.

**MNIST**  The MNIST dataset [23] is a widely used benchmark consisting of a total of 70,000 grayscale images of handwritten digits (0-9). The dataset is split into a training set of 60,000 images and a testing set of 10,000 images. It comprises 10 distinct classes, with each class containing approximately 7,000 images in total (6,000 for training and 1,000 for testing). Each image has a resolution of $28 \times 28$ pixels.

**Fashion-MNIST**  Fashion-MNIST [46] is designed as a direct replacement for the original MNIST, offering a more challenging benchmark with images of clothing items. This dataset also contains 70,000 grayscale images ($28 \times 28$ pixels), split into 60,000 for training and 10,000 for testing. It features 10 distinct classes of apparel, with a similar distribution of approximately 7,000 images per class.

**CIFAR-10**  The CIFAR-10 dataset [21] consists of 60,000 color images ($32 \times 32$ pixels) categorized into 10 distinct classes representing real-world objects such as animals, vehicles, and fruits. The dataset is typically divided into 50,000 training images and 10,000 testing images, with 6,000 images per class evenly split between the training and testing sets (5,000 for training, 1,000 for testing).

**CIFAR-100** CIFAR-100 [21] is a finer-grained classification dataset containing 60,000 color images ($32 \times 32$ pixels), typically split into 50,000 for training and 10,000 for testing. It features 100 distinct classes, with each class containing exactly 600 images (500 for training and 100 for testing). These classes can also be grouped into 20 broader superclasses. The images depict a wide variety of objects, providing a more challenging and granular classification task than CIFAR-10.

**Omniglot** For exploring the discovery of character hierarchies, we utilized the Omniglot dataset [22]. Omniglot is a collection of 1,623 different handwritten characters from 50 alphabets. Each character was drawn by 20 different individuals, resulting in 20 samples per class. This dataset is characterized by a large number of classes and a small number of samples per class, making it suitable for evaluating the model's ability to form hierarchies in a few-shot setting. The images are typically grayscale. For our experiments, we exclusively used the training set of the Omniglot dataset.

## D Additional Implementation Details

### D.1 Encoder-Decoder Architecture

The encoder architecture varies according to input image type and dimensions:

**Grayscale Datasets** ($28 \times 28$ **pixels**). For datasets such as MNIST and FashionMNIST, the encoder applies three sequential convolutional layers to shrink the spatial dimensions from $28 \times 28$ down to $3 \times 3$ while increasing channel capacity:

- **Conv1:** kernel $3 \times 3$, stride 2, padding 1, maps $1 \times 28 \times 28 \rightarrow 8 \times 14 \times 14$, followed by Batch Normalization [14] and ReLU.
- **Conv2:** kernel $3 \times 3$, stride 2, padding 1, maps $8 \times 14 \times 14 \rightarrow 16 \times 7 \times 7$, followed by Batch Normalization and ReLU.
- **Conv3:** kernel $3 \times 3$, stride 2, padding 0, maps $16 \times 7 \times 7 \rightarrow 32 \times 3 \times 3$, followed by Batch Normalization and ReLU.

The resulting $32 \times 3 \times 3$ tensor serves as the encoded feature representation. For Omniglot, the encoder comprises six sequential convolutional layers:

- **Conv1:** $3 \times 3$, stride 1, padding 1, $1 \rightarrow 32$, preserves $28 \times 28$, + BatchNorm + ReLU.
- **Conv2:** $4 \times 4$, stride 2, padding 0, $32 \rightarrow 32$, down to $13 \times 13$, + BatchNorm + ReLU.
- **Conv3:** $3 \times 3$, stride 1, padding 1, $32 \rightarrow 64$, maintains $13 \times 13$, + BatchNorm + ReLU.
- **Conv4:** $4 \times 4$, stride 2, padding 0, $64 \rightarrow 64$, down to $5 \times 5$, + BatchNorm + ReLU.
- **Conv5:** $3 \times 3$, stride 1, padding 1, $64 \rightarrow 128$, maintains $5 \times 5$, + BatchNorm + ReLU.
- **Conv6:** $4 \times 4$, stride 2, padding 0, $128 \rightarrow 128$, down to $1 \times 1$, followed by ReLU.

**RGB Datasets** ($32 \times 32$ **pixels**). For datasets such as CIFAR-10 and CIFAR-100, the encoder employs modified ResNet-style blocks [12] used in [29]. Each residual block includes a weighted skip connection scaled by 0.1. The number of convolutional filters starts from 32 and doubles with each downsampling stage, culminating in a final feature map size of $256 \times 4 \times 4$.

**Latent Space and Decoder.** The latent representation $\mathbf{z}$ has a dimension of 8 for grayscale datasets and 64 for RGB datasets. The decoder mirrors the encoder architecture, utilizing transposed convolutional layers to reconstruct the input images from the latent vector.

### D.2 Regularizer Terms

To learn a stable taxonomic hierarchy on a large amount of clusters (i.e., a large $\mathcal{T}$), we propose two regularizers to help stabilize the training. The first regularizer penalizes trivial parent splits such that one of the two children has a very low convex weight $\alpha$. Following [8], we impose an entropy regularizer on $\alpha$ at each branch to encourage a uniform split. We decay the entropy regularizer weight exponentially towards the leaf with $\lambda_{\text{ent}}$. Formally,

$$\mathcal{R}_{\text{ent}}(\mathcal{T}) = \sum_{c \in \mathcal{T}} \lambda_{\text{ent}}^{\text{depth}(c)} \Big[ -\alpha_c \log \alpha_c - (1 - \alpha_c) \log(1 - \alpha_c) \Big]$$

Additionally, in a large taxonomic hierarchy, leaf-level clusters tend to be close to each other, as fewer discriminative features can be used to separate two children down the tree. To encourage discovering as much taxonomic prototypes as possible, the second regularizer penalizes any two leaf clusters from being too close together by a margin set by hyperparameter as measured by their symmetric KL divergence. We decrease the weighting exponentially bottom-up with $\lambda_{\mathrm{dkl}}$. Formally,

$$\mathcal{R}_{\mathrm{dkl}}(\mathcal{T}) = \sum_{c_{\mathrm{left}}, c_{\mathrm{right}} \in \mathcal{T}} \max\Big\{ 0,\ m\, \lambda_{\mathrm{dkl}}^{N-\mathrm{depth}(c)} - \Big[ D_{\mathrm{KL}}\big(c_{\mathrm{left}} \,\|\, c_{\mathrm{right}}\big) + D_{\mathrm{KL}}\big(c_{\mathrm{right}} \,\|\, c_{\mathrm{left}}\big) \Big] \Big\}$$

where $m$ is the margin and $N$ is the depth of $\mathcal{T}$. In out experiment, we set $m$ to 1.2 and both $\lambda_{\mathrm{ent}}$ and $\lambda_{\mathrm{dkl}}$ to 0.01. We refer to Appendix E.2 for ablation studies on the two regularizer terms.

### D.3 Full List of Hyperparameter

We provide the list of hyperparameter used in our experiment in Table 4.

| Hyperparameter | Value |
| --- | --- |
| **Training** | |
| Learning rate | $1 \times 10^{-3}$ |
| Batch size | 256 |
| Epochs | 400 |
| $|\mathcal{T}|$ | 10 layers, or 2047 clusters |
| $\lambda_{\mathrm{dkl}}$ | 0.01 |
| $\lambda_{\mathrm{ent}}$ | 0.01 |
| $m$ | 1.2 |
| **ELBO Loss** | |
| Reconstruction weight | 5.0 |
| KL divergence weights | 1.0 |
| **Contrastive Loss** | |
| Embedding contrastive temperature | 0.5 |
| Clustering contrastive temperature | 0.3 |
| Loss weight | 100 |
| **Model** | |
| `dim(z)` | 8 for grayscale image, 64 for RGB |

Table 4: Summary of hyperparameter settings.

### D.4 Compute Resources

Experiments are conducted on a single NVIDIA A40 GPU. Estimated training times are clearly provided for reproducibility:

- MNIST, FashionMNIST, and Omniglot training typically requires approximately 45 minutes per run.
- CIFAR-10 and CIFAR-100 require approximately 3 hours per training run.

## E Ablation Study

### E.1 Contrastive Learning

To understand the impact of the two contrastive loss terms, we conduct an ablation study presented in Table 5. We evaluate four configurations: Embedding only (embedding-level contrastive loss), Clustering only (clustering-level contrastive loss), No contrastive (no contrastive learning), and the

Full model (both contrastive terms applied). The results show that any form of contrastive learning improves performance over the baseline, with the full model achieving the highest scores across all four hierarchical clustering metrics.

| Ablations | DP | LP | ACC | NMI |
|---|---|---|---|---|
| Full | 42.74 | 54.81 | 67.13 | 51.34 |
| Clustering only | 39.60 | 48.84 | 65.41 | 50.08 |
| Embedding only | 22.34 | 38.10 | 40.92 | 21.91 |
| No contrastive | 19.60 | 37.05 | 38.59 | 20.07 |

Table 5: Ablation study of contrastive loss on CIFAR-10.

## E.2 Regularizers

To evaluate the impact of the regularizer terms, we conduct an ablation study on both Fashion and CIFAR-10 datasets, as shown in Tables 6 and 7. We analyze four settings: the Full model (both regularizers applied), $\mathcal{R}_{ent}$ only, $\mathcal{R}_{dkl}$ only, and No regularizers.

The results indicate that the entropy regularizer ($\mathcal{R}_{ent}$) significantly improves the hierarchical purity metrics—DP and LP—especially LP, across both datasets. Notably, on CIFAR-10, the combination of both regularizers achieves the best performance across all four metrics, suggesting that the two terms are complementary in optimizing the hierarchical structure.

In contrast, on the Fashion dataset, the effect of both regularizers is less pronounced, particularly for the hierarchical clustering accuracy metrics—ACC and NMI. This difference may stem from the higher complexity and greater intra- and inter-class variance of CIFAR-10, where the regularizers contribute more effectively to refining hierarchical boundaries.

Overall, our results suggest that $\mathcal{R}_{ent}$ is crucial for enhancing hierarchical purity, while the combination of $\mathcal{R}_{ent}$ and $\mathcal{R}_{dkl}$ proves especially effective in managing more complex datasets like CIFAR-10.

| Ablations | DP | LP | ACC | NMI |
|---|---|---|---|---|
| Full | 42.74 | 54.81 | 67.13 | 51.34 |
| $\mathcal{R}_{ent}$ only | 42.13 | 52.85 | 66.98 | 50.70 |
| $\mathcal{R}_{dkl}$ only | 41.43 | 50.92 | 66.02 | 49.98 |
| No regularizers | 40.38 | 50.26 | 65.39 | 48.17 |

Table 6: Ablation study of regularizer terms on CIFAR-10.

| Ablations | DP | LP | ACC | NMI |
|---|---|---|---|---|
| Full | 59.12 | 81.44 | 81.10 | 72.29 |
| $\mathcal{R}_{ent}$ only | 58.91 | 81.01 | 81.04 | 72.29 |
| $\mathcal{R}_{dkl}$ only | 54.69 | 78.71 | 80.11 | 72.12 |
| No regularizers | 52.60 | 78.30 | 80.62 | 72.39 |

Table 7: Ablation study of regularizer terms on Fashion.

# F  Broader Impacts and Risks

## F.1  Broader Impacts

We introduces Deep Taxonomic Networks for unsupervised hierarchical prototype discovery, a method inspired by the human cognitive capacity for learning, organizing knowledge, and forming

hierarchical conceptual structures[42][35][9][2], particularly the principles of hierarchical taxonomies and prototype representation[35].

### F.1.1 Positive Impacts

By automatically organizing complex, unlabeled data into interpretable hierarchical taxonomies and revealing associated prototypes[39], our model can significantly improve data exploration and understanding. It can provide more interpretable representations compared to flat clustering methods[25][28]. This method will be valuable in biology, material science and social science, where discovering inherent structures in large datasets leads to new insights, hypothesis generation, and innovative discovery.

Inspired by human hierarchical concept formation[42] and the psychological relevance of basic-level categories[4] [16], this work could potentially contribute to developing more intuitive and effective educational tools or interfaces that help users organize and understand complex information by visually representing hierarchical relationships, building on computational models of categorization like Cobweb[1][13].

### F.1.2 Negative Impacts

As an unsupervised learning method, deep taxonomic networks are susceptible to learning and potentially amplifying biases present in the training data. If the data reflects societal biases (e.g., in representation of certain demographic groups or concepts), the learned taxonomic structure and prototypes could entrench these biases, potentially leading to unfair or discriminatory outcomes if the model is used in downstream applications that affect individuals or groups.

### F.2 Risks

Given that the model is a deep generative latent variable model within the VAE framework [14][18][36], it learns to model the data distribution. If applied to data that could be used to generate or organize misleading information, such as grouping images or text in a biased way, the discovered hierarchies could potentially be exploited to make fake content appear more structured or credible, contributing to the spread of disinformation. While the model operates on unlabeled data, the discovery of fine-grained prototypes and hierarchical clusters across different levels of the taxonomy could potentially reveal sensitive or private information, particularly if the discovered categories are highly specific or linkable to individuals.

## G   Licenses

We provide details regarding the licenses of external assets used in this work, presented in Table 8.

| Asset | URL | License |
|-------|-----|---------|
| **Datasets** | | |
| MNIST [23] | Link | Creative Commons Attribution-Share Alike 3.0 |
| Fashion-MNIST [46] | Link | MIT License |
| CIFAR-10/100 [21] | Link | MIT License |
| Omniglot [22] | Link | Creative Commons Attribution-ShareAlike 4.0 International License |
| **Code** | | |
| Tree VAE Code [29] | Link | MIT License |

Table 8: Licenses for assets used in our experiments.

