# OpenReview forum: "Deep Taxonomic Networks for Unsupervised Hierarchical Prototype Discovery"
_NeurIPS.cc/2025/Conference — NeurIPS 2025 poster_

### Official Review · Reviewer_MkCw · 2025-06-27

**Clarity:** 3
**Significance:** 2
**Originality:** 3
**Rating:** 5
**Confidence:** 4

**Summary:**

The authors introduce a novel deep hierarchical clustering algorithm, where cluster hierarchies are defined through learned prototypes. This approach is conceptually interesting and demonstrates a degree of novelty. However, the empirical validation is currently limited. In particular, the paper lacks strong baselines and evaluations on complex image datasets, which hinders a thorough assessment of the method's effectiveness [1]. Incorporating additional deep flat clustering methods and testing on more challenging datasets would significantly strengthen the experimental section and better highlight the practical significance and robustness of the proposed approach.

[1] Lu, Yiding, et al. "A survey on deep clustering: from the prior perspective." Vicinagearth 1.1 (2024): 4.

**Questions:**

Your experimental section includes comparisons with TreeVAE and DeepECT. However, it omits more recent and advanced deep clustering methods, such as SCAN. Could you clarify  the rationale behind excluding these methods from your comparison, and  how your approach performs when applied to more complex patterns, such as those found in  STIL-10 or Image-net?

**Ethical Concerns:**

["NO or VERY MINOR ethics concerns only"]

**Final Justification:**

The paper is technically solid, and my major concerns regarding the experimental section have been largely resolved.

**Limitations:**

The central promise and selling ("deep taxonomic networks discover rich and interpretable hierarchical taxonomies, capturing both coarse-grained semantic categories and fine-grained visual distinctions") would need more substantiation.
Furthermore, the evaluation lacks recent and stronger baselines such as SCAN, and does not address performance on complex datasets like  STIL-10 or Image-net, leaving the method’s  competitiveness unclear.

**Paper Formatting Concerns:**

-

**Quality:**

3

**Strengths And Weaknesses:**

The technique is somewhat new in the context of deep clustering. It is theoretically/statistically well founded. I believe that the central promise and selling ("deep taxonomic networks discover rich and interpretable hierarchical taxonomies, capturing both coarse-grained semantic categories and fine-grained visual distinctions") would need more substantiation.

Although the proposed prototype-based hierarchical clustering framework is conceptually novel, but its empirical validation is limited. The absence of strong baselines and the use of only simple datasets hinder a clear assessment of the method's effectiveness. Moreover, the lack of comparison to related approaches such as deep flat clustering further weakens the evaluation. Including more competitive methods and complex datasets would significantly improve the work’s credibility.

---

> ### Author Rebuttal · Authors · 2025-07-31
>
> Thank you for your review. We appreciate that you found our hierarchical clustering approach novel and interesting and that our theoretical treatments seem well founded.
>
> We would like to address your main concerns:
>
> ### **Concerns on the lack of evaluation on more complex patterns such as STL-10:**
>
> - ***“...the use of only simple datasets hinder a clear assessment of the method's effectiveness…Including more … complex datasets would significantly improve the work’s credibility…how your approach performs when applied to more complex patterns, such as those found in STIL-10…”***
>     - We chose the  evaluation datasets and the encoder-decoder architecture based on the prior literature on variational approaches for hierarchical clustering (TreeVAE, NeurIPS 2023), so we could make a direct comparison with prior work. We believe the selected datasets cover a wide range of characteristics (i.e., grayscale vs. RGB images, handwritten vs. natural images, simple vs. complicated patterns, and 10 classes vs 100 classes) and are sufficiently comprehensive for our **proof-of-concept** study. **Nevertheless, we agree that inclusion of results on a more complex dataset would be more convincing, and we not that reviewer Lzud also made a similar suggestion.**
>     - **Based on these suggestions, we will add additional results for STL-10 (a popular subset of ImageNet-1k), which we were able to produce  during the short rebuttal period given our available compute resources. The results show that our approach is also robust in the face of more complex patterns:**
>
>
>         | **STL-10 Results** |  |  |  |  |
>         | --- | --- | --- | --- | --- |
>         |  | DP | LP | ACC | NMI |
>         | TreeVAE | 20.34% | 31.31% | 30.96% | 25.74% |
>         | **DeepTaxonNet** | **29.03%** | **37.93%** | **55.75%** | **40.56%** |
>
> ### **Concerns on the lack of comparison to deep flat clustering methods such as SCAN:**
>
> - ***“...omits more recent and advanced deep clustering methods, such as SCAN…Could you clarify the rationale behind excluding these methods from your comparison…?”:***
>     - We would like to first clarify the scope of our work. We frame our work as a **proof-of-concept** experiment to show that our novel variational approach to hierarchical clustering **can discover hierarchical prototypes from data**. As a result, our analysis is centered around the added value of our approach (utilizing all levels of the hierarchy vs. other hierarchical clustering methods that do not learn intermediate prototypes). We used 4 hierarchical clustering metrics (LP, DP, ACC, NMI) to evaluate multiple hierarchical clustering methods and **our methods' benefit comes from the collective analysis from these metrics.** For instance,
>         - Leaf Purity (LP) measures the class entropy at the leaf nodes, and in our case are the nodes representing the most fine-grain semantic categories. As stated in our response to **reviewer Ludz**, Table 1 shows that despite having many more leaves ($2^{10}$ leaves) than TreeVAE and DeepECT (10 leaves, corresponding to 10 classes), our approach's leaf purity is comparable to other approaches (our approach has even higher LP), suggesting it forms **robust**, **consistent**, and **well-structured** fine-grained semantic classes at the **leaves**.
>         - *An interesting observation from the leaf-only approaches is that ACC is lower than LP*. This is expected as leaf-only approaches assume the same number of leaves as the classes so the classification accuracy depends on both the quality of leaf nodes representing a class (LP) and the routing quality that successfully brings data to a leaf node and hence *ACC should be upper bounded by LP*. **However, with our approach, we observe a substantial increase in ACC (e.g., on CIFAR datasets) compared to TreeVAE, despite having a similar LP. Notably, ACC is not upper-bounded by LP, as is the case with leaf-only approaches.** This result indicates that the *additional* ACC gain in our approach is the benefit of utilizing intermediate nodes. **In other words, by utilizing all levels, our approach can capture in the intermediate nodes what would otherwise be misclassified in the leaf nodes of leaf-only approaches. This translates into a better classification accuracy**.
>     - However, deep flat clustering methods lack hierarchical metrics such as LP and DP, and the only comparable metrics are ACC and NMI. As a result, results on deep flat clustering methods fail to provide a comprehensive analysis to our approach’s added value.
> - ***“...the evaluation lacks recent and stronger baselines such as SCAN,...”:***
>     - While we feel that comparing our new approach to deep flat clustering approaches is beyond the scope of our current work, we agree that showing competitiveness in *classification accuracy* to a stronger approach would significantly improve our work’s credibility. Based on this idea, we ran an analysis of *classification accuracy* between our approach and SCAN on CIFAR-10 for your reference.
>     - Our approach utilized the same encoder-decoder architecture as TreeVAE, so that we could directly compare to this prior approach. However, this encoder-decoder architecture is less powerful than a ResNet-18. To enable a direct comparison to SCAN, we utilized the ResNet-18 encoder-decoder architecture within our approach and tested it on CIFAR-10. For this experiment we kept the same hyperparameters as in our previous experiment. **We found that our approach achieves a much higher classification accuracy on CIFAR-10 when using the ResNet-18 encoder-decoder architecture as compared to the TreeVAE encoder-decoder architecture (\~77\% vs. \~68\% currently). With this updated architecture, our approach achieves similar performance to SCAN without self-labeling (\~81.8\%), even though we have not yet  tuned the hyperparameters for our approach with this new encoder-decoder architecture**.
>     - This result suggests that our theoretical framework indeed benefits from a stronger encoder-decoder architecture and is competitive in classification accuracy to a stronger approach like SCAN. While we did not evaluate our approach with the ResNet-18 encoder-decoder on the other datasets (we did not have sufficient time during the rebuttal period), **we plan to include an additional ablation study on the effect of stronger encoder-decoder architectures as well as *performance comparison* to approach like SCAN.**
>
> We would like to thank you again for your valuable review and we believe the additional results for STL-10 and for CIFAR-10 with the updated encoder-decoder architecture add more credibility to our experimental validation and substantially strengthen the paper.

---

> > ### Comment · Reviewer_MkCw · 2025-08-03
> >
> > Thank you for the detailed rebuttal. While your additional STL-10 and SCAN comparisons are helpful, key issues—especially broader baselines and large-scale scalability—remain insufficiently resolved. Accordingly, I will keep my original score.

---

> ### Author Response · Authors · 2025-08-03
>
> We appreciate your reply. Given the short rebuttal period and limited compute, we added STL-10 results and a SCAN comparison (matched backbone) as requested. These results indicate **scalability** and **competitive performance** beyond the original scope. We agree that broader baselines and larger-scale experiments will strengthen the paper. **These runs are already in progress**, and we will include: **1) direct performance comparisons to recent flat clustering approaches**, and **2) results on ImageNet-1k.** We hope this further clarifies our rebuttal.
>
> To resolve the remaining concern, could you kindly specify which additional baselines and which large-scale dataset you consider essential so we can target those directly?

---

> ### Comment · Reviewer_MkCw · 2025-08-05
>
> Thank you very much for your response.
> I absolutely understand the advantage of hierarchical clustering compared to flat clustering.
> However, in the end, the user often wants the final clustering in order to work with it directly — that's why I also asked about state-of-the-art deep clustering methods.
> Since 2023, countless methods have been published, but these are not reflected in the experiments.
> Overall, I think the work is good, but the experimental  section still needs structural development.
>
> Datasets: Food101 dataset, INaturalist21 dataset
>
> Flat clustering: Lu, Yiding, et al. "A survey on deep clustering: from the prior perspective." Vicinagearth 1.1 (2024): 4.
>
> H-Clustering: Palumbo, Emanuele, et al. "From logits to hierarchies: Hierarchical clustering made simple." arXiv preprint arXiv:2410.07858 (2024). and accepted in ICML 2025

---

> > ### Author Response · Authors · 2025-08-06
> >
> > We thank you for providing additional clarifications. We would like to argue that our approach is indeed comparable to recent SOTA methods (Palumbo et al., 2025), given similar image encoders. Specifically, we train our approach directly using DINOv2-giant encodings, as in Palumbo et al. (2025), on CIFAR-10/100 and Food-101. All hyperparameters are kept the same as in our previous experiments. Below is the classification accuracy comparison:
> > | **Accuracy** | CIFAR-10 | CIFAR-100 | Food-101 |
> > | --- | --- | --- | --- |
> > | L2H-TEMI | 95.6 | 68.2 | **90.4** |
> > | L2H-Turtle | **99.5** | **89.6** | 87.6 |
> > | **DeepTaxonNet** | **99.1** | **88.7** | 84.1 |
> >
> > Our approach matches the classification accuracy reported by Palumbo et al. (2025), while additionally offering much deeper hierarchical clusters, including intermediate clusters, and without requiring prior knowledge of the number of labels. We believe that even better performance can be achieved by experimenting with different learning rates, learning rate schedulers, DTN depths, and latent dimensions, especially for more complex datasets like Food-101.
> >
> > As a result, we propose the following revision plan for the final version:
> > - Further clarifies our main goal/contribution as raised by reviewer Lzud and NMPg;
> > - We will include an additional section on “plug-and-play” pre-trained features for challenging datasets such as Food-101, iNaturalist21, and ImageNet-1k. We will compare our approach against SOTA flat and hierarchical clustering methods in terms of classification performance.
> > - The purpose is to support the claim that our approach can benefit from stronger feature representations and can scale up to demonstrate robustness.

---

> > > ### Comment · Reviewer_MkCw · 2025-08-06
> > >
> > > I thank the authors for their detailed and thoughtful responses. Most of my concerns have been addressed, and I appreciate the inclusion of the additional results. I have therefore updated my rating accordingly.

---

> > > > ### Author Response · Authors · 2025-08-06
> > > >
> > > > We are glad we could address your concerns! Thank you once again for your valuable feedback!

---

### Official Review · Reviewer_b5Xg · 2025-06-30

**Clarity:** 3
**Significance:** 2
**Originality:** 3
**Rating:** 4
**Confidence:** 4

**Summary:**

This paper proposes a deep hierarchical clustering method - deep taxonomic networks (DeepTaxonNet).  In contrast to existing methods such as DeepECT and TreeVAE, DeepTaxonNet puts more emphasis on the intermediate nodes, allowing clusters associated with every node in the tree to be a potential source for generating an observation.  The tree structure is defined as a complete binary tree with an associated hierarchical mixture of Gaussians prior.  Distributions at parent nodes are constrained to be a convex combination of its children's distributions.  Variational inference is performed by maximizing the Evidence Lower Bound, and a derivation is given that shows that maximizing this bound encourages maximization of Categorical Utility.  Comparison is given on several data sets with the existing methods of Cobweb, DeepECT, and TreeVAE, where the proposed method gives better performance using the proposed metrics in the paper that account for the explicit use of intermediate nodes.  Additionally, DeepTaxonNet allows for adaptation to a new data set without re-training, and is shown to give better results on this task compared with retraining TreeVAE.

**Questions:**

Please see the weaknesses section above.  With regard to that section, would it be fair to say the main advantage of the proposed approach and leveraging the intermediate nodes is that the level of granularity does not need to be pre-defined, and so the same learned taxonomy can be used for varying tasks without re-training?  Are there additional benefits that you see beyond this?

**Ethical Concerns:**

["NO or VERY MINOR ethics concerns only"]

**Final Justification:**

Taking into account the other reviews and the author feedback and updated results, I will maintain my original rating.

**Limitations:**

yes

**Quality:**

3

**Strengths And Weaknesses:**

strengths:

- evaluation with existing methods on multiple data sets, ablation study given in appendix, demonstration of adaptation to new task without retraining

weaknesses:

- it seems the main contribution/novelty of the proposed method is the more explicit use of non-leaf nodes as having their own generating distributions and serving as potential prototypes.  and this comes at the cost of having a rigid pre-defined binary tree structure.
- in that regard then, I would have liked to see more analysis of the benefits of this approach.  in part because the evaluation metrics have been adapted to reflect the use of intermediate nodes, so from a comparison standpoint are not completely fair to the existing methods that are only designed to use the leaf nodes.
- for instance, new task classification without re-training seems like an advantage, but it would be nice to see more experiments along those lines
- the qualitative analysis is 5.3 is interesting, but there is not an explicit comparison with existing methods, and in any case more quantitative analysis of the benefits of using the intermediate nodes would be more convincing.

---

> ### Author Rebuttal · Authors · 2025-07-31
>
> Thank you for your detailed review. We are glad you found our experiment and ablation study comprehensive and that you are interested in our method’s ability to adapt to new tasks without requiring retraining. Below are point-by-point responses and clarifications to your specific concerns.
>
> - **“*...I would have liked to see more analysis of the benefits of this approach. in part because the evaluation metrics have been adapted to reflect the use of intermediate nodes, so from a comparison standpoint are not completely fair to the existing methods that are only designed to use the leaf nodes…*”:**
>     - Thank you for raising this concern. As **reviewer Lzud** pointed out, we believe more clarification/analysis of our approach’s benefits will strengthen our argument. To the best of our knowledge, **we are the first to use a variational approach to learn hierarchical generative distributions at all levels of the tree**. Given our use of a tree structure and a variational approach, we thought TreeVAE (NeurIPS 2023) was the most recent and relevant baseline for our approach. While it might seem “unfair” to compare to methods that only use the leaf node, **we argue that the use of these intermediate nodes is a key contribution of our approach over prior approaches**. The 4 hierarchical clustering metrics we chose collectively provide a complete analysis at both leaf- and whole tree-level:
>         - Leaf Purity (LP) measures the class entropy at the leaves. In our approach, these nodes represent the most fine-grain semantic classes. Table 1 shows that despite having many more leaf nodes ($2^{10}$ leaves) than TreeVAE and DeepECT (10 leaves, corresponding to 10 classes), our approach's leaf purity is comparable to the others (in a sense that we have even higher LP). This suggests our approach forms **robust**, **consistent**, and **well-structured** fine-grained semantic classes at the **leaf level**.
>         - *An interesting observation from the leaf-only approaches is that ACC is lower than LP*. This is expected as leaf-only approaches assume the same number of leaves as the classes so the classification accuracy depends on both the quality of leaf nodes representing a class (LP) and the routing quality that successfully brings data to a leaf node and hence *ACC should be upper bounded by LP*. **However, with our approach, we observe a substantial increase in ACC (e.g., on CIFAR datasets) compared to TreeVAE, despite having a similar LP. Notably, ACC is not upper-bounded by LP, as is the case with the leaf-only approach.** This result indicates that the *additional* ACC gain in our approach is the benefit of utilizing intermediate nodes. As we mentioned in our response to **Reviewer Ludz, by utilizing all levels, our approach can capture in intermediate nodes what would otherwise be misclassified in the leaf nodes of leaf-only approaches, producing a better classification accuracy**.
>     - Again, we view our approach's ability to utilize all levels of the hierarchy as an **advantage and a novel research contribution** instead of an “unfair comparison to baselines”.
> - ***“...new task classification without re-training seems like an advantage, but it would be nice to see more experiments along those lines…”:***
>     - We appreciate that you found our approach's ability to perform new task classification without re-training advantageous. We are particularly excited about the benefits of this capability in cases where there are large training costs that might prohibit re-training.
>     - One possible direction for future work is to test our framework via large scale training on a real-world dataset, such as applying it to ImageNet-21k data to learn a large hierarchy. We hope to compare the classification performance of our approach on new datasets without re-training against more traditional approaches that use distinct pre-training and fine-tuning stages. Our current work, however, aims to first provide a **proof-of-concept** that our novel variational approach **can successfully learn hierarchical prototypes from data**. Given the theoretical and empirical results presented in the current work, we hypothesize that **one of the practical benefits of our approach will lie in compute- and data-efficient unsupervised and zero-shot classification**.
>     - Because our approach learns generative clusters at the intermediate nodes and subsequently supports classification on a new task without re-training, there is no directly comparable baselines to perform new task classification without re-training in the deep hierarchical clustering literature (that use intermediate nodes) . Most existing approaches involve pre-training and fine-tuning (i.e., SimCLR, MAE) that are first pre-trained on large scale dataset (ImageNet) and then fine-tuning on the new tasks. We believe these other approaches are qualitatively different than the approach we propose as well as the line of prior approaches we use as baselines (i.e., TreeVAE, DeepECT). Our work centered around theoretical inquiries and empirical validation of proposed framework, and we worry that including fine-tuning approaches in our paper will make the proposed framework less clear.
> - ***“...the qualitative analysis is 5.3 is interesting, but there is not an explicit comparison with existing methods…”:***
>     - We also appreciate that you find our qualitative analysis interesting. We want to point out that the purpose of Section 5.3 is to showcase that our approach's learned hierarchy, including intermediate nodes, is both consistent and human interpretable, while the hierarchies learned with previous approaches are worse on both accounts.
> - **“*...and in any case more quantitative analysis of the benefits of using the intermediate nodes would be more convincing…*”:**
>     - Given your suggestion (**and those of the other reviewers**), we plan to add additional results to our paper. Specifically, we plan to further validate our approach’s effectiveness in utilizing intermediate nodes for more complex images. We provide a comparison of our approach and TreeVAE on the STL-10 dataset, a popular subset of the ImageNet:
>
>
>         | **STL-10 Results** |  |  |  |  |
>         | --- | --- | --- | --- | --- |
>         |  | DP | LP | ACC | NMI |
>         | TreeVAE | 20.34% | 31.31% | 30.96% | 25.74% |
>         | **DeepTaxonNet** | **29.03%** | **37.93%** | **55.75%** | **40.56%** |
>     - Additionally, **reviewer MkCw** called for a comparison to SCAN which uses a ResNet-18 architecture and to make our approach comparable (i.e., to control for the encoder-decoder), we did an evaluation using this same encoder-decoder architecture within our approach. Interestingly, we found that the new encoder-decoder improved our system in the **CIFAR-10 case (~77\% vs. ~68\% currently).** While we did not evaluate the other datasets using this architecture because we did not have sufficient time during the rebuttal period, **we plan to include an additional ablation study on the effect of stronger encoder-decoder architectures**.
>     - These results collectively suggest that our theoretical framework indeed benefits from a stronger encoder-decoder and that even more complex patterns (STL-10) it can achieve robust performance. This provides more evidence to the idea that utilization of intermediate nodes can translate into better predictions, as suggested above.
>
> - ***“…would it be fair to say the main advantage of the proposed approach and leveraging the intermediate nodes is that the level of granularity does not need to be pre-defined, and so the same learned taxonomy can be used for varying tasks without re-training? Are there additional benefits that you see beyond this?…”:***
>     - We agree that using the taxonomy learned from one task, without the level of granularity being pre-defined, to support classification on another without re-training is a key benefit of using intermediate nodes. We also see several additional benefits, such as:
>         - *Utilizing intermediate nodes benefits the classification performance* as our approach can capture in intermediate nodes what would otherwise be misclassified in the leaf nodes of leaf-only approaches (Table 1, section 5.2).
>         - *Utilizing intermediate nodes facilitates the discovery of taxonomic structures within unlabeled datasets (section 5.3).* Our approach can structure, organize, and partition data in-the-wild (in a sense we have no prior knowledge about its semantic categories) for us to analyze and interpret.
>
> We would like to thank you again for your constructive reviews. We will do our best to incorporate your concerns and suggestions into the final version. We believe the  final paper—updated based on your feedback—will offer a more insightful analysis of the benefits of our approach and strengthen the credibility of our quantitative results.

---

> > ### Author Response · Authors · 2025-08-06
> >
> > We would like to inform you that we have added new experimental results demonstrating the scalability and competitiveness of our approach with stronger feature representations (DINOv2) during discussion with Reviewer MkCw. These updates can be found under the thread for Reviewer MkCw. We are looking forward to your feedback.

---

### Official Review · Reviewer_NMPg · 2025-07-01

**Clarity:** 2
**Significance:** 2
**Originality:** 2
**Rating:** 3
**Confidence:** 3

**Summary:**

This paper presents *Deep Taxonomic Networks* (DTN), a novel deep latent variable framework for unsupervised hierarchical prototype discovery. The method introduces a mixture-of-Gaussians prior over a complete binary tree to model hierarchical abstractions and leverages variational inference jointly with contrastive learning to learn taxonomic structures from unlabeled data.

**Questions:**

Please refer to Weakness.

**Ethical Concerns:**

["NO or VERY MINOR ethics concerns only"]

**Limitations:**

Please refer to Weakness.

**Quality:**

3

**Strengths And Weaknesses:**

**Strengths:**

1. **Strong theoretical foundation**: The method section is rigorous, and the derivation of the ELBO objective is comprehensive. In particular, the reinterpretation of ELBO as a prototypicality maximization objective (Section 3.3) provides an insightful information-theoretic justification.

2. **Well-structured model design**: The use of a binary-tree-structured prior enables the discovery of hierarchical abstractions without assuming the number of ground-truth classes, which is a desirable property for unsupervised clustering.

3. **Interpretability**: The model discovers visually coherent and semantically meaningful sub-hierarchies, which enhances interpretability beyond flat clustering methods.


**Weaknesses and Suggestions:**

1. **Outdated baselines**: The experimental comparison is mainly limited to older methods such as Cobweb, TreeVAE, and DeepECT. These baselines, while relevant historically, do not fully reflect the current state of unsupervised and self-supervised clustering, especially in the context of vision transformers or modern contrastive learning models. Recent approaches should be included for a more compelling comparison.

2. **Limited dataset diversity**: The evaluation is conducted on standard benchmarks like MNIST, Fashion-MNIST, CIFAR-10/20. These datasets, while widely used, are relatively outdated and limited in semantic complexity. They may not fully demonstrate the benefits of discovering deep taxonomic structures. I recommend evaluating on more recent and semantically structured datasets.

3. **Missing ablation analysis**: While the paper introduces several important components, I could not find any dedicated ablation study isolating the contributions of these individual components. As a result, it is difficult to assess which aspects of the method are most responsible for the observed performance gains. I strongly recommend including ablation experiments to disentangle the effects of the hierarchical prior, contrastive learning, and other design choices. Additionally, although the supplementary materials were available for download, I was unable to open them successfully; it is possible that relevant analysis was included there, but I could not verify it.

---

> ### Comment · Reviewer_NMPg · 2025-08-06
>
> The authors did not address my rebuttal. My concern remains unresolved, and I will maintain my original score.

---

> > ### Author Response · Authors · 2025-08-07
> >
> > We thank your time for the review.
> >
> > 1 and 2: We want to argue that TreeVAE (NIPS 2023) is not an outdated baseline but the most recent VAE approach to hierarchical clustering. However, we agree that showing more competitiveness to the state-of-the-arts approach will make our approach more robust. Hence, as other reviewer suggested, we first ran experiments on STL-10 using our default settings and encoder-decoders:
> > | **STL-10 Results** |  |  |  |  |
> > | --- | --- | --- | --- | --- |
> > |  | DP | LP | ACC | NMI |
> > | TreeVAE | 20.34% | 31.31% | 30.96% | 25.74% |
> > | **DeepTaxonNet** | **29.03%** | **37.93%** | **55.75%** | **40.56%** |
> >
> > Second, we would like to argue that our approach is indeed comparable to recent SOTA methods, L2H (Palumbo et al., 2025), given similar image encoders. Specifically, we train our approach directly using DINOv2-giant encodings, as in Palumbo et al. (2025), on CIFAR-10/100 and Food-101 (101 classes, 101,000 images, 512x512 resolution). All hyperparameters are kept the same as in our previous experiments. Our approach matches the performance reported by Palumbo et al. (2025), while additionally offering much deeper hierarchical clusters, including intermediate clusters, and without requiring prior knowledge of the number of labels. We believe that even better performance can be achieved by experimenting with different learning rates, learning rate schedulers, DTN depths, and latent dimensions, especially for more complex datasets like Food-101.
> > | **MNI** | CIFAR-10 | CIFAR-100 | Food-101 |
> > | --- | --- | --- | --- |
> > | L2H-TEMI | 90.1 | 77.8 | **91.7** |
> > | L2H-Turtle | **98.5** | **91.7** | 89.4 |
> > | **DeepTaxonNet** | **97.4** | **89.4** | 83.5 |
> >
> > | **Accuracy** | CIFAR-10 | CIFAR-100 | Food-101 |
> > | --- | --- | --- | --- |
> > | L2H-TEMI | 95.6 | 68.2 | **90.4** |
> > | L2H-Turtle | **99.5** | **89.6** | 87.6 |
> > | **DeepTaxonNet** | **99.1** | **88.7** | 84.1 |
> >
> > | **LP** | CIFAR-10 | CIFAR-100 | Food-101 |
> > | --- | --- | --- | --- |
> > | L2H-TEMI | 95.8 | 69.8 | 88.1 |
> > | L2H-Turtle | 99.5 | 89.6 | 84.3 |
> > | **DeepTaxonNet** | **99.6** | **93.0** | **90.1** |
> >
> > | **DP** | CIFAR-10 | CIFAR-100 | Food-101 |
> > | --- | --- | --- | --- |
> > | L2H-TEMI | 90.2 | 50.2 | 80.1 |
> > | L2H-Turtle | **98.8** | **80.3** | **75.8** |
> > | **DeepTaxonNet** | 88.0 | 71.0 | 70.3 |
> >
> > We argue that we choose a simpler encoder-decoder to learn a less powerful image representation for the sake of direct comparison to prior work. However, as noted above, performance increases dramatically when using a more powerful image representation, demonstrating its partical use.
> >
> > 3: Appendix E (pp. 18–19) presents ablation settings on contrastive and regularization terms and referenced in line 186 in the main text. Additionally, reviewer b5Xg claims our detailed ablation study as one of the strengths. Regarding the file error, what were the specific error messages that you ran into? We were happy to further clarifies if you can provide additional information.
> >
> > As a result, we propose the following revision plan for the final version:
> >
> > - Further clarifies our main goal/contribution;
> > - We will include an additional section (i.e. page 10) on “plug-and-play” pre-trained features for challenging datasets such as Food-101, iNaturalist21, and ImageNet-1k. We will compare our approach against SOTA flat and hierarchical clustering methods in terms of classification performance.
> > - The purpose is to support the claim that our approach can benefit from stronger feature representations for practical use and can scale up to demonstrate robustness.
> >
> > Palumbo, et al. From logits to hierarchies: Hierarchical clustering made simple.

---

### Official Review · Reviewer_Lzud · 2025-07-05

**Clarity:** 2
**Significance:** 3
**Originality:** 3
**Rating:** 4
**Confidence:** 3

**Summary:**

The paper proposes a new approach for hierarchical classification, i.e., instead of predicting "collie" for an image, a hierarchical classification would predict a chain animal --> mammal --> dog --> collie. The paper proposes to relax the condition of every node in the hierarchy tree to correspond to conventional semantic classes and associate them instead to clusters of samples (images). If I understood correctly, the paper claims to learn both the structure of the hierarchy as well as the classification technique. Each node in the tree is represented by a latent variable that is learned using a variational approach. The paper derives the expected lower bound to maximize for this problem and use encoder-decoder structure to learn the latents and predict classes.

The proposed method was tested on public datasets CIFAR, MNIST and Fashion datasets.

**Questions:**

..

**Ethical Concerns:**

["NO or VERY MINOR ethics concerns only"]

**Final Justification:**

See my comment below.

**Limitations:**

..

**Paper Formatting Concerns:**

..

**Quality:**

3

**Strengths And Weaknesses:**

Strengths:
1. Attempt to mathematically formulate the overall problem will be appreciated in the community. (Pls refer to other reviewers on method novelty).
2. In several places, the writeup attempts to provide interpretation of mathematical quantities connecting them to the problem setting -- I think this is important for paper with heavy theoretic treatments. I also appreciate the effort on good amount of experimental details, results and analysis performed in the paper.

Weaknesses:
1. Goal/Contribution: I dont think I understood the merit of the proposed approach. If the tree leaves do not represent any semantic class, I dont understand their meaning, or more importantly, their benefit. Would these more freeform clusters at leaf advantageous for classification? The accuracies on Table 1 does not support that. Clarification on what is the benefit of this structure is needed for understanding the value of this contribution.
Same concern can extended to internal nodes as well. From the writeup, if appears the clusters pertaining to internal nodes are not exclusive at the same level. What benefit does this form of internal nodes offers us?

2. Method: Does the method learns the hierarchy structure or assumes it is provided? I assumed it does, but not completely sure on it. Lines 188~200 appear to suggest a hierarchy needs to be provided in the beginning but are not clear whether or not this is updated. If it does, this should be clarified in Section 3 how the structure is learned in the variational approach.

The method section also could elaborate a bit further how the $\alpha_i$'s (introduced in Line 119) are learned by the encoder-decoder design. Line 169 briefly mentions it, could help to elaborate. In general, would help to include a paragraph how the quantities needed for the hierarchical prediction (e.g., as shown in Figure 2) are learned through encoder-decoder, i.e., connecting the mathematical formulation to implementation.

3. Experimental validation:
3a. The datasets (CIFAR, MNIST) selected for evaluation are a bit experimental and simpler datasets. In order to convince the reader the robustness of this method, results on more real life datasets, at least ImageNet or OpenImages, are necessary.
3b. The proposed method seems to be achieving good performance on the simpler MNIST dataset. Given the DP and LP in 40s and 20s for CIFAR, and accuracy be 70%, I cannot help questioning the goal again as to how the mixed clusters at leaves and internal nodes are beneficial .

---

> ### Author Rebuttal · Authors · 2025-07-31
>
> Thank you for your detailed review and substantive feedback. We were glad to hear that you found our mathematical formulation sound, our writing approachable, and our experiments and analyses substantial.
>
> ### **Clarification of the scope:**
>
> Before responding to your concerns, we would like to first clarify the scope of our work. We frame our work as a **proof-of-concept** experiment to show that our novel variational approach to hierarchical clustering **can discover hierarchical prototypes from data**. To this end, we first *analytically* show that optimizing the lower bound of our approach encourages the discovery of prototypical nodes across the hierarchy given input data (as supported by section 3.3). Then, we *empirically* show that the most prototypical nodes given some inputs indeed happen at *different levels* across the hierarchy, **including the leaf nodes**.
>
> ### **For concerns on goal/contribution (“*...Clarification on what is the benefit of this structure is needed for understanding the value of this contribution…*”):**
>
> - **“*...I dont think I understood the merit of the proposed approach. If the tree leaves do not represent any semantic class, I dont understand their meaning, or more importantly, their benefit….*”:**
>     - Our main objective and contribution is that **we assume nodes at *any* level can represent semantic (taxonomic) classes,** including leaf nodes. While the terminology of “leaf” and “intermediate” come from the tree data structure, we do not mean to make a distinction between them with respect to semantic categories. Leaf nodes (lower-level nodes)  represent  “finer-grained” semantic classes than their parents, rather than being  “freeform” or “not represent[ing] any semantic class[es]”.
> - ***“...Would these more freeform clusters at leaf advantageous for classification? The accuracies on Table 1 does not support that…”:***
>     - Leaf Purity (LP) measures the class entropy at the leaves (the nodes with the most fine-grained semantic classes). **Table 1** shows that despite having substantially more leaf nodes ($2^{10}$ nodes) than TreeVAE and DeepECT (10 nodes, corresponding to 10 classes), our approach's leaf purity is comparable to the baseline approaches (it has higher LP), suggesting **robust**, **consistent**, and **well-structured** fine-grained semantic classes at the **leaf level**. In other words, while other approaches have 10 leaves corresponding to the 10 classes, ours can identify more subclasses that are inherent in the data, but are not explicitly called out in the labelings. This is beneficial for classification because our approach enables the model to better disentangle subtle semantic differences that might found similar across labels (e.g., a digit ‘4’ that is similar to a ‘9' in MNIST, or an ‘automobile’ that is similar to a ‘truck’ in CIFAR-10).
> - ***“...What benefit does this form of internal nodes offers us?...”:***
>     - *An interesting observation from the leaf-only approaches (TreeVAE and DeepECT) is that ACC is lower than LP.* This is expected as leaf-only approaches assume the same number of leaves as the classes so the classification accuracy depends on both the quality of leaf nodes representing a class (LP) and the routing quality that successfully brings data to a leaf node and hence *ACC should be upper bounded by LP*. **However, with our approach, we observe a substantial increase in ACC (e.g., on CIFARs) compared to TreeVAE, despite having a similar LP. Notably, ACC is not upper-bounded by LP, as is the case with the leaf-only approach.** This result indicates that the *additional* ACC gain in our approach is a benefit of utilizing additional intermediate nodes. In other words, by utilizing all levels, our approach can correctly capture in intermediate nodes what would otherwise be misclassified in the leaf nodes of leaf-only approaches, results in better classification accuracy.
>
> ### **For concerns on method:**
>
> - ***“...Does the method learns the hierarchy structure or assumes it is provided? I assumed it does, but not completely sure on it….”:***
>     - **The method learns the semantics of the hierarchy from the data.**  We assume that the hierarchical structure is a binary tree, but do not provide any hierarchical semantic information. In other words, we pre-allocate a semantically empty structure prior to learning. All semantic categories are learned/discovered through unsupervised training.
> - **“*...Lines 188~200 appear to suggest a hierarchy needs to be provided in the beginning but are not clear whether or not this is updated….*”:**
>     - Lines 188-200 describes the *inference* mechanism employed **after** a hierarchy has been learned (line 191-194). We describe inference first, so the reader can understand how the hierarchy will ultimately be used before describing how it is learned. Prior to inference, we train the hierarchy from unlabeled data—we do not provide any semantic information about the hierarchical categories. During inference (lines 188-200), we perform a forward pass on the *learned* hierarchy to map discovered hierarchical semantic information to the actual class labels of interest for evaluation, as the model is trained without labels, we need to provide labels afterwards to enable classification.
> - ***(From Summary) “...I understood correctly, the paper claims to learn both the structure of the hierarchy as well as the classification technique….”:***
>     - Yes, we *learn* the hierarchical categories and after we map them to the provided  classification labels, we can use them to make predictions.
> - ***“...The method section also could elaborate a bit further how the \alpha's (introduced in Line 119) are learned by the encoder-decoder design… connecting the mathematical formulation to implementation…”:***
>     - Our approach learns by updating parameters outlined in Figure 2. Parent nodes (as parameterized by $\mu_{c_\{parent}}$, and $\sigma^2_{c_\{parent}}$) are computed via $\alpha_c$ using the equation from Section 3.1. During back propagation, we compute the gradient for the $\alpha_c$ parameters and use these gradients to adjust them accordingly to minimize the overall loss.
>
> ### **For concerns on experimental validation:**
>
> - ***“...selected for evaluation are a bit experimental and simpler datasets. In order to convince the reader the robustness of this method, results on more real life datasets…”:***
>     - We use the selected dataset and encoder-decoder architecture following prior literature on variational approach to hierarchical clustering (TreeVAE, NeurIPS 2023) for a direct comparison. We believe the selected datasets vary across a wide range of relevant characteristics (i.e., grayscale vs. RGB images, handwritten vs. natural images, simple vs. complex patterns, and 10 classes vs 100 classes), making them comprehensive for our **proof-of-concept** study.
>     - Nevertheless, we agree that inclusion of results for a more real life dataset could better convince readers of the robustness of our approach. **In alignment with your feedback and that of  reviewer MkCw, we will add the following results for STL-10, a popular subset of ImageNet-1k.** We were able to evaluate our approach and TreeVAE on this dataset during the short rebuttal period given our compute resources**.** Here are the results we plan to add, which highlight our approach’s robust performance on more complex datasets:
>         |  | DP | LP | ACC | NMI |
>         | --- | --- | --- | --- | --- |
>         | TreeVAE | 20.34% | 31.31% | 30.96% | 25.74% |
>         | **DeepTaxonNet** | **29.03%** | **37.93%** | **55.75%** | **40.56%** |
>     - Additionally, **reviewer MkCw** called for a comparison to SCAN which uses a ResNet-18 architecture. And to make our approach comparable (i.e., controlling for the encoder-decoder), we did an evaluation using this other architecture. Interestingly, we found that the new encoder-decoder improved our system in the **CIFAR-10 cases (~77\% vs. ~68\% currently).** While we did not evaluate the other datasets using this architecture because we did not have sufficient time during the rebuttal period, **we plan to include an additional ablation study on the effect of stronger encoder-decoder architectures**. These collective results suggest that our theoretical framework indeed benefits from a stronger encoder-decoder and that it is has robust performance, even with more complex patterns (STL-10).
> - **“*...Given the DP and LP in 40s and 20s for CIFAR, and accuracy be 70%, I cannot help questioning the goal again as to how the mixed clusters at leaves and internal nodes are beneficial…*”:**
>     - We want to clarify that our approach's DP is ~43\%, LP ~54\%, ACC ~68\% on CIFAR-10 and DP ~17\%, LP ~28\%, ACC ~41\% on CIFAR-20, because CIFAR-20 contains less samples per class and hence is more challenging.
>     - **Our goal is to learn taxonomies from data, such that a model can discover finer and finer semantic clusterings down the hierarchy, without requiring prior information about a dataset’s semantic labelings.** To achieve this, we proposed a novel variational approach to hierarchical clustering that encodes data at multiple levels of granularity. And given our approach’s ability to learn hierarchical semantic clusterings, our approach can:
>         1. Capture in intermediate nodes what would otherwise be misclassified in the leaf nodes of leaf-only approaches. This lets our approach achieve better classification accuracy.
>         2. Identify subclasses that are inherent in the data (as will be reflected in the lower level (leaf level) of the hierarchy), but are not explicitly called out in the data labelings.
>
> We thank you again for your thoughtful review and the concerns you raised. We believe the changes based on your feedback substantially strengthen our paper and increase the  contribution and credibility of our work.

---

> > ### Author Response · Authors · 2025-08-06
> >
> > We would like to inform you that we have added new experimental results demonstrating the scalability and competitiveness of our approach with stronger feature representations (DINOv2) during discussion with Reviewer MkCw. These updates can be found under the thread for Reviewer MkCw. We are looking forward to your feedback.

---

> ### Comment · Reviewer_Lzud · 2025-08-06
>
> I appreciate the authors' effort to address the questions and concerns.
>
> After reading the rebuttal, my main concerns that are not addressed are as follows:
>
> 1. The experiments were performed on simpler datasets, I see additional experiments on STL-10, which is a small subset of imagenet.
> 2. The tree structure learning is still not clear. Instead of deriving the equations on the main paper, the text should clarify how the structure is learned to maintain semantic meaning. The rebuttal also talks about inference mostly, instead it would be better if the training process was clarified.
> 3. With low LPs and DPs, what would the user of this approach gain from this method? I see the accuracies improved for multiple dataset to match the existing models, but, we can achieve those from methods without the hierarchical structure. What is the practical benefit of the hierarchical modeling?
>
> I am increasing the rating, but I am not fully convinced this work would attract a lot of attention in its current form. I would strongly recommend resubmitting to another venue after restructuring the paper with enough details of structure learning and clearly explaining the benefit of the proposed hierarchical approach, and perhaps explore avenues to achieve high DP with matching SOTA classification accuracy.

---

> ### Author Response · Authors · 2025-08-06
>
> We thank your feedback and appreciate that you increased our ratings.
> 1. We have added new experimental results demonstrating the scalability and competitiveness of our approach with stronger feature representations (DINOv2) during discussion with Reviewer MkCw. We would like to argue that our approach is indeed comparable to recent SOTA methods, L2H (Palumbo et al., 2025), given similar image encoders. Specifically, we train our approach directly using DINOv2-giant encodings, as in Palumbo et al. (2025), on CIFAR-10/100 and Food-101 (101 classes, 101,000 images, 512x512 resolution). All hyperparameters are kept the same as in our previous experiments. Our approach matches the performance reported by Palumbo et al. (2025), while additionally offering much deeper hierarchical clusters, including intermediate clusters, and without requiring prior knowledge of the number of labels. We believe that even better performance can be achieved by experimenting with different learning rates, learning rate schedulers, DTN depths, and latent dimensions, especially for more complex datasets like Food-101. Below is the comparison to SOTA:
> | **MNI** | CIFAR-10 | CIFAR-100 | Food-101 |
> | --- | --- | --- | --- |
> | L2H-TEMI | 90.1 | 77.8 | **91.7** |
> | L2H-Turtle | **98.5** | **91.7** | 89.4 |
> | **DeepTaxonNet** | **97.4** | **89.4** | 83.5 |
> | --- | --- | --- | --- |
> | **Accuracy** | **CIFAR-10** | **CIFAR-100** | **Food-101** |
> | L2H-TEMI | 95.6 | 68.2 | **90.4** |
> | L2H-Turtle | **99.5** | **89.6** | 87.6 |
> | **DeepTaxonNet** | **99.1** | **88.7** | 84.1 |
> | --- | --- | --- | --- |
> | **LP** | **CIFAR-10** | **CIFAR-100** | **Food-101** |
> | L2H-TEMI | 95.8 | 69.8 | 88.1 |
> | L2H-Turtle | 99.5 | 89.6 | 84.3 |
> | **DeepTaxonNet** | **99.6** | **93.0** | **90.1** |
> | --- | --- | --- | --- |
> | **DP** | **CIFAR-10** | **CIFAR-100** | **Food-101** |
> | L2H-TEMI | 90.2 | 50.2 | 80.1 |
> | L2H-Turtle | **98.8** | **80.3** | **75.8** |
> | **DeepTaxonNet** | 88.0 | 71.0 | 70.3 |
>
> 2. The tree structure is pre-defined (randomly initialized) by our hierarchical Gaussian prior and learned by optimizing the ELBO as described in section 3.2. Section 3.3 showed that our approach is also optimizing the mutual information between data's latent representation and our hierarchical Gaussian prior. An intutive way to think of the learning is to consider each node in the tree as a bin that clusters data. In a standard VAE, where the prior is a standard Gaussian, there is only 1 bin (with $\mu=0$, $\sigma^2=1$) for all data. Consider a modified VAE where we have multiple Gaussian priors. By optimizing the ELBO of this modifed VAE, we are learning different Gaussian prior that generates the data, where each prior now is a bin that clusters the data. Let's consider a modified VAE with these multiple Gaussian priors, but now having a hierarchical constraints on them where parents are the weighted sum (the learnable $\alpha$) of their children, so parents always enclose the semantic classes of their children--our approach. So optimizing our ELBO is jointly updating the $\alpha$ and the different Gaussian priors. However, it will NOT learn a freeform semantic structure (i.e. things -> fruits and cars -> apple). Section 3.3 showed that optimizing ELBO of our approach encourages each Gaussian prior to contain as pure features as possible (higher mutual information, i.e. vehicles -> cars -> Ford cars). And since parents always enclose the semantic classes of their children, it would discourage the case where a Gaussian prior contains mixed semantic classes, resulting a structured hierarchical prior.
>
> 3. We argue that we choose a simpler encoder-decoder to learn a less powerful image representation for the sake of direct comparison to prior work. However, as noted in **1.**, performance increases dramatically when using a more powerful image representation, demonstrating its partical use. The practical benefit of the hierarchical modeling, particularly our approach, is that we don't need prior knowledge of the number of class for clustering. Our approach also enables the model to better disentangle subtle semantic differences that might found similar across labels (e.g., a digit ‘4’ that is similar to a ‘9' in MNIST, or an ‘automobile’ that is similar to a ‘truck’ in CIFAR-10) in higher level nodes.
>
> As a result, we propose the following revision plan for the final version:
>
> - Further clarifies our main goal/contribution;
> - We will include an additional section (i.e. page 10) on “plug-and-play” pre-trained features for challenging datasets such as Food-101, iNaturalist21, and ImageNet-1k. We will compare our approach against SOTA flat and hierarchical clustering methods in terms of classification performance.
> - The purpose is to support the claim that our approach can benefit from stronger feature representations for practical use and can scale up to demonstrate robustness.
>
> Palumbo, et al. From logits to hierarchies: Hierarchical clustering made simple.

---

### Note · Authors · 2025-08-12

## Scope & method
Our goal is a proof-of-concept that a variational framework with a **fixed-topology binary-tree** mixture-of-Gaussians prior can learn hierarchical semantics from data (structure is architectural; semantics are learned). Optimizing the ELBO jointly updates node parameters and mixing weights and can be viewed as increasing mutual information between data latents and the hierarchical prior—yielding purer, nested prototypes rather than free-form clusters. We clarified this training-time mechanism (as opposed to inference) in discussion.

## Why intermediate nodes matter
Using all levels gives practical benefits beyond leaf-only methods:
1. Intermediate prototypes recover cases that would otherwise be misrouted to imperfect leaves (i.e., leaves mixing multiple classes), improving accuracy.
2. The same learned taxonomy serves multiple levels of granularity and supports new classification tasks without retraining.
3. Interpretability via prototype paths.

We summarized these points during discussion.

## Datasets & baselines
We followed recent hierarchical VAE work (TreeVAE, NeurIPS’23) to ensure a direct comparison, spanning grayscale/RGB, simple/complex patterns, and 10–100 classes, appropriate for a proof-of-concept, while acknowledging the value of richer datasets during discussion:
- **Complex data**: On STL-10 we substantially outperform TreeVAE demonstrating robustness on more challenging images.
- **Competitiveness with SOTA under strong features**: With DINOv2-giant features (as in L2H/Palumbo et al., 2025), our method is competitive on CIFAR-10/100 and Food-101 with higher Leaf Purity while still yielding deeper, interpretable hierarchies and requiring no prior on label counts.

This new evidence indicates that our approach scales competitively and further underscores the value of intermediate nodes, strengthening our overall contribution and credibility (addressing Reviewer Lzud's remaining concerns).

## Planned revisions
- Further clarifies our scope and benefits of intermediate nodes.
- Add a “plug-and-play pre-trained features” section (Food-101, iNat21, ImageNet-1k) to show competitiveness and robustness of our approach.


Palumbo, et al. From logits to hierarchies: Hierarchical clustering made simple.

---

### Decision · Program_Chairs · 2025-09-17

**Decision:**

Accept (poster)

**Comment:**

This paper proposes Deep Taxonomic Networks (DTN), which uses variational inference and contrastive learning to discover flexible taxonomies from unlabeled data. The approach introduces a mixture-of-Gaussians prior structured over a complete binary tree, with both leaf and intermediate nodes representing clusters/prototypes.

The paper received scores of 4, 3, 4, 5. Most reviewers appreciated the theoretical overview and the mathematical formulation of the problem. There were questions on the benefit of intermediate clustering and the limited evaluations. However, through the rebuttal phase the authors have clarified several questions about comparison to baselines and evaluations on more recent benchmarks, which has convinced the reviewers about the validity and usefulness of this method. Thus, we recommend the paper for an Accept.